# Leader360V: A Large-scale, Real-world 360 Video Dataset for Multi-task Learning in Diverse Environments

**Weiming Zhang**[1*]    **Dingwen Xiao**[1*]    **Aobotao Dai**[1*]    **Yexin Liu**[2]
**Tianbo Pan**[3]    **Shiqi Wen**[1]    **Lei Chen**[1,2]    **Lin Wang**[4† *]

[1] HKUST (GZ)    [2] HKUST    [3] National University of Singapore
[4] Nanyang Technological University

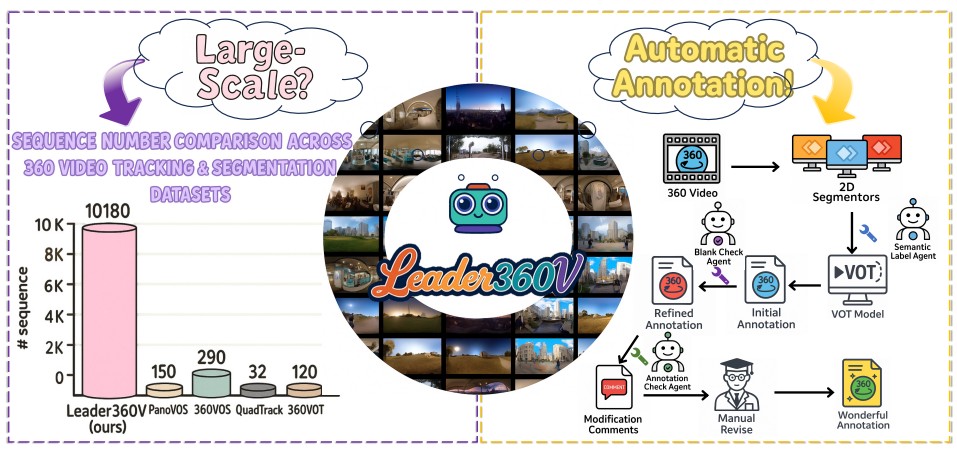

Figure 1: The overall of our Leader360V dataset.

## Abstract

360 video captures the complete surrounding scenes with the ultra-large field of view of 360×180. This makes 360 scene understanding tasks, *e.g.*, segmentation and tracking, crucial for appications, such as autonomous driving, robotics. With the recent emergence of foundation models, the community is, however, impeded by the lack of large-scale, labelled real-world datasets. This is caused by the inherent spherical properties, *e.g.*, severe distortion in polar regions, and content discontinuities, rendering the annotation costly yet complex. This paper introduces **Leader360V**, the **first** large-scale (10K+), labeled real-world 360 video datasets for instance segmentation and tracking. Our datasets enjoy high scene diversity, ranging from indoor and urban settings to natural and dynamic outdoor scenes. To automate annotation, we design an automatic labeling pipeline, which subtly coordinates pre-trained 2D segmentors and large language models (LLMs) to facilitate the labeling. The pipeline operates in three novel stages. Specifically, in the **Initial Annotation Phase**, we introduce a Semantic- and Distortion-aware Refinement (**SDR**) module, which combines object mask proposals from multiple 2D segmentors with LLM-verified semantic labels. These are then converted into mask prompts to guide SAM2 in generating distortion-aware masks for subsequent frames. In the **Auto-Refine Annotation Phase**, missing or incomplete regions are corrected either by applying the SDR again or resolving the discontinuities near the horizontal borders. The **Manual Revision Phase** finally incorporates LLMs and human annotators to further refine and validate the annotations. Ex-

---

[*]Equal contribution. [†]Corresponding author.

39th Conference on Neural Information Processing Systems (NeurIPS 2025) Track on Datasets and Benchmarks.

tensive user studies and evaluations demonstrate the effectiveness of our labeling pipeline. Meanwhile, experiments confirm that Leader360V significantly enhances model performance for 360 video segmentation and tracking, paving the way for more scalable 360 scene understanding. We release our dataset and code at https://leader360v.github.io/Leader360V_HomePage/ for better understanding.

# 1 Introduction

360 cameras, *a.k.a*, panoramic cameras, provide an ultra-large field of view (FoV) of $360 \times 180$ for the surrounding environment. Therefore, 360 video enables comprehensive situational awareness that surpasses the FoV limitations of perspective 2D cameras. This makes 360 camera-based scene understanding popular and crucial for applications such as autonomous driving [1, 2, 3, 4, 5], robotics [6, 7], and virtual reality [8, 9]. A commonly used representation for 360 videos is the equirectangular projection (ERP), which maps the spherical content onto a 2D rectangular plane to ensure compatibility with the standard imaging pipeline. However, ERP poses several challenges specific to 360 video, including projection distortions in polar regions, and horizontal discontinuities [5] that break content continuity across the left and right borders. These challenges significantly increase the cost and complexity of manual annotation for 360 videos.

Although several 360 video benchmarks [5, 10, 11] have been proposed for scene understanding tasks, such as segmentation and tracking, the scale and diversity of these datasets remain limited by far in the community, especially with the recent emergence of foundation models [12]. For segmentation, 360VOS [10] contains 290 panoramic sequences annotated across 62 categories, while PanoVOS [5] provides 150 high-resolution videos with instance masks. For the tracking task, 360VOT [13] focuses on single-object tracking with 120 omnidirectional videos covering 32 object types, and QuadTrack [11] introduces a small-scale multi-object tracking benchmark under non-uniform motion. However, the modest size and task-specific design of these datasets limit their ability to support large-scale, generalizable model training. In contrast, recent 2D video benchmarks such as YouTube-VOS [14], with over 3,400 videos and 540K segmentation annotations, and SA-V [12], with 50.9K videos and over 640K masklets, have demonstrated the value of dense, large-scale annotation for training foundation models.

This disparity raises a key scientific question: *Can we build a large-scale 360 video dataset with rich annotations for both segmentation and tracking tasks, while substantially reducing the human labeling cost?* In this paper, we present **Leader360V** (Sec. 3.1), the first large-scale (10K+), real-world 360 video dataset with dense, frame-level annotations for scene understanding tasks across segmentation and tracking. Leader360V covers 198 object types and covers a wide variety of scenes, including both indoor and outdoor environments, as shown in Fig. 2. Leader360V is constructed by integrating existing public datasets with our self-collected 360 videos captured in diverse real-world environments, yielding a scalable and representative benchmark for panoramic understanding.

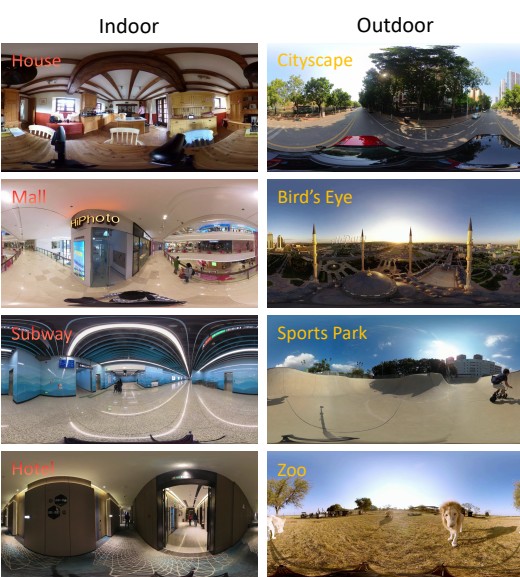

Figure 2: Samples from different scenarios of the Leader360V dataset.

To enable the construction of Leader360V, we also propose $A^3$360V (Automatic Annotate Any 360 Video)(Sec. 3.2), a novel annotation pipeline tailored for 360 videos. $A^3$360V is designed to reduce manual labeling burden while maintaining high annotation quality through a three-phase pipeline: **Initial Annotation Phase**(Sec. 3.2.1): We first extract keyframes and use multiple 2D segmentors (e.g., CropFormer [15], OneFormer [16]) to generate semantic and instance segmentation proposals. These outputs are unified and aligned via LLM-based semantic matching, then refined through a Semantic- and Distortion-aware Refinement (SDR) Module that leverages

Table 1: **Comparison of 360 video datasets on segmentation (VOS) and tracking (VOT).** **"Mobile"**: videos shot with motion. **"Still"**: videos shot without any motion. **"Vehicle"**: videos shot on vehicles. **"Human"**: videos shot by humans while walking or running. **"Attr"**: characteristics of tracking (Single-Object and Multi-Object Tracking) and segmentation (Partial Frame and Whole Frame Segmentation). **"Auto"**: no human involvement except revise. **"Manu"**: no assistant model involvement.

| Task | Dataset | Vol | State | | Foundation | | Avg | Class | Attr | Anno |
|---|---|---|---|---|---|---|---|---|---|---|
| | | | Mobile | Still | Vehicle | Human | | | | |
| | 360VOT [10] | 120 | 96 | 24 | 89 | 7 | 940f | 32 | SOT | Semi |
| | PanoVOS [5] | 150 | 21 | 129 | 13 | 8 | 20s | 35 | SOT | Semi |
| 360VOT | QuadTrack [11] | 32 | 32 | 0 | 32 | 0 | 60s | N/A | MOT | Manual |
| | JRDB [17] | 54 | 32 | 22 | 32 | 0 | 70s | N/A | MOT | Manual |
| | Leader360-T (Ours) | 10180 | 5K+ | 5K+ | 2K+ | 3K+ | 15s | 198 | MOT | Auto |
| | 360VOS [10] | 170 | 135 | 35 | 124 | 11 | 940f | 32 | PFS | Semi |
| | PanoVOS [5] | 150 | 21 | 129 | 13 | 8 | 20s | 35 | PFS | Semi |
| 360VOS | WOD [18] | 1150 | 1150 | 0 | 1150 | 0 | 20s | 28 | WFS | Manual |
| | Leader360-S (Ours) | 10180 | 5K+ | 5K+ | 2K+ | 3K+ | 15s | 198 | WFS | Auto |

SAM2 to produce high-quality panoramic masks. **Auto-Refine Annotation Phase**(Sec. 3.2.2): For subsequent keyframes in the video, we iteratively propagate annotations and identify low-quality regions based on mask coverage. Frames failing coverage thresholds are reprocessed using a GPT-guided Motion-Continuity Refinement (MCR) module, which resolves annotation inconsistencies across left-right ERP and recovers missing masks caused by occlusion or distortion. **Manual Revise Phase**(Sec. 3.2.3): Finally, human annotators validate and correct the outputs from the previous stages. Multi-annotator review ensures consistency and completeness across frames, producing the final high-quality annotations.

Extensive validation confirms the effectiveness of our pipeline. User studies show that $A^3360V$ significantly reduces annotator workload while preserving annotation quality. Experiments on standard 360 video segmentation and tracking benchmarks demonstrate that Leader360V enhances model performance, paving the way for robust, scalable, and generalizable 360 video understanding.

In summary, our contributions are three-fold: (**I**) We propose **Leader360V**, the first large-scale (10K+), labeled real-world 360 video dataset specifically designed for instance segmentation and tracking in diverse and dynamic environments. (**II**) We also propose $A^3$**360V** (Automatic Annotate Any 360 Video) pipeline, which integrates pre-trained 2D segmentors with large language models to automate the annotation process and significantly reduce human effort without compromising label quality. (**III**) Extensive user studies and experimental results validate the effectiveness of our Lead360V and proposed pipeline and highlight the potential of Leader360V to advance robust 360 video understanding.

## 2 Related Works

**Video-based panoramic datasets for object tracking and segmentation.** 360 video, with its omnidirectional coverage, offers advantages over conventional 2D video, such as a broader field of view, richer spatial context, and greater understanding of the continuous scene. These benefits have led to the development of various 360 video datasets across different tasks, including object tracking [13, 11, 17], and segmentation [5, 10, 18]. Object tracking in 360 videos has been explored through single-object and multi-object tracking benchmarks. For instance, 360VOT [13] provides the first dataset for omnidirectional single-object tracking, while QuadTrack [11] captures non-uniform motion using a quadruped robot to establish a multi-object tracking challenge. Segmentation, which is more annotation-intensive, demands pixel-level masks and is mainly represented by datasets focused on instance and panoptic segmentation, such as PanoVOS [5] and 360VOS [10]. These datasets help address 360-specific challenges such as distortion and content continuity. However, most existing datasets remain limited in scale and task diversity, restricting their ability to support robust and generalizable learning. *To this end, we introduce Leader360V, a large-scale 360 video dataset constructed by integrating publicly available resources and newly self-collected videos, enhanced by an automatic annotation pipeline for multi-task learning. Detailed comparison is shown in Tab. 1.*

**Automated Annotation Frameworks for Scalable Dataset Construction.** As large-scale video datasets continue to grow, the demand for efficient annotation has led to the emergence of semi-automatic and automatic pipelines aimed at reducing manual labeling costs. For the annotation of 360 video, it is a complex task that necessitates specialized attention due to its unique characteristics, such as severe distortion, a wide field of view, and discontinuous context across panoramic borders. 360 video annotation methods such as 360Rank [19] and PanoVOS [5] adopt semi-supervised pipelines using pre-trained segmentors and keyframe propagation, but still rely heavily on manual mask drawing and semantic labeling, limiting scalability in 360 settings. However, these methods are not essentially different from 2D video annotation strategies [12, 20, 21] and do not take into account the special characteristics of 360 videos. To address these gaps, we learn from recent automatic annotation systems [22, 23], which have incorporated large language models (LLMs) to further reduce human involvement. *We propose $A^3360V$, a unified annotation framework tailored for 360 videos. By integrating LLMs for semantic role assignment and pre-trained 2D segmentors for initial mask generation, $A^3360V$ enables scalable segmentation and tracking from keyframes to full video sequences under omnidirectional conditions.*

**Large-Scale 2D Video Segmentation and Tracking Datasets.** Compared to the challenges faced in constructing large-scale 360 video datasets, the field of 2D video understanding has witnessed the emergence of numerous large-scale datasets for segmentation and tracking tasks. YouTube-VOS [14], LVOS [20], MeViS [24], VIPSeg [21], SA-V [12] for segmentation task and TrackingNet [25], LaSOT [26], TAO [27] for tracking task have a large base or long shots of a single video, the largest of which can exceed 50K, and a single video can exceed 7 hours. *Motivated by the gap between the rapid expansion of 2D video resources and the limited availability of large-scale 360 video datasets, we introduce Leader360V, a richly annotated 360 video dataset designed for segmentation and tracking tasks.*

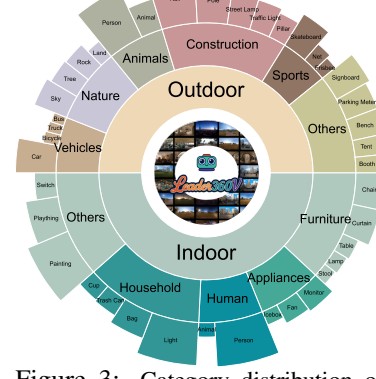

Figure 3: Category distribution of Leader360V dataset.

**Higher Diversity in Scenarios, Especially Cityscape.** Existing 360 video datasets provide limited coverage of cityscape scenarios. This gap hinders the development of practical 360VOTS applications in real-world urban environments. Therefore, we prioritize the inclusion of a wide range of urban environments in Leader360V, capturing variations in architectural styles, traffic conditions, and other dynamic elements. Our Leader360V is also rich in categories, as shown in Fig. 3

## 3 Methodology

### 3.1 The Leader360V Dataset

To address the scarcity of large-scale 360 video datasets, we present **Leader360V**—the first real-world dataset of this scale with diverse scene dynamics and comprehensive annotations for instance segmentation (Leader360V-S) and tracking (Leader360V-T).

### 3.1.1 Date Source Analysis

The Leader360V dataset includes videos collected from existing 360 video datasets, such as 360VOTS [10], PanoVOS [5], *etc*. The specific information is shown in Tab. 2. To address the limited scene diversity in prior datasets, we additionally collecte new videos and relabel existing videos. Departing from the collection protocols used in previous 360 video datasets, our self-collected videos exhibit three distinctive properties, as described below.

**Richer Data Acquisition Methods.** We employ a variety of recording techniques to capture

Table 2: **Our Data Source**. **"Pct"**: percentage of selected data. **"Sel"**: specific number of selected data. VG: Video Generation. VC: Video Caption

| Source* | Task | Pct | Sel | Relabel |
|---|---|---|---|---|
| 360VOTS* [10] | 360VOT | 80% | 232 | ✓ |
| PanoVOS* [5] | 360VOS | 60% | 90 | ✓ |
| WEB360* [28] | 360 VG | 50% | 1K+ | ✓ |
| 360+x* [8] | 360 VC | 30% | 1K+ | ✓ |
| YouTube360* [29] | 360 VC | 20% | 3K+ | ✓ |
| Open Source | N/A | N/A | 1K+ | ✓ |
| Self-Collected | 360VOTS | N/A | 2K+ | ✓ |

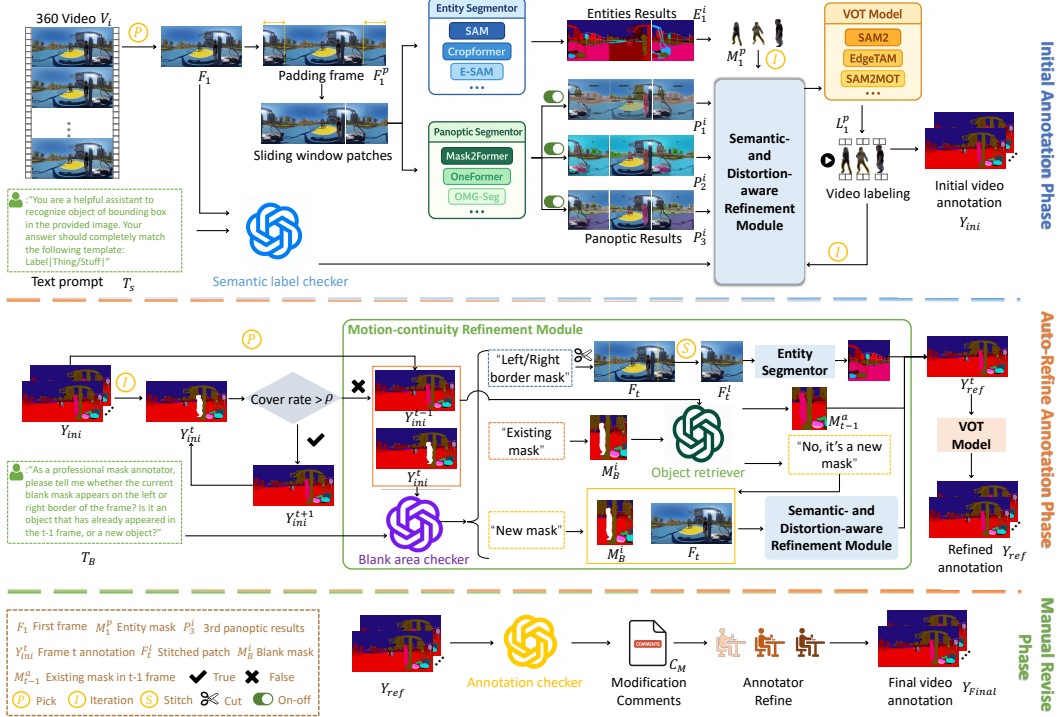

Figure 4: The overall of our A³360V pipeline. It consists of three phases: Initial Annotation Phase, Auto-Refine Annotation Phase, and Manual Revise Phase.

diverse camera motion patterns, including static camera setups, handheld recordings by moving photographer, and vehicle-based capture. In contrast to previous datasets that rely on limited recording methods, our approach enriches the diversity of 360 videos by simulating a wider range of real-world camera movements, as shown in Tab. 1.

**More Various Perspectives.** We collect data from multiple viewpoints and perspectives within each scenario. For example, in vehicle-based videos, we include footage from both the roof and the side of the car. This multi-perspective collection is often ignored by previous works.

### 3.1.2 Pre-Processing

All videos in our Leader360V, whether sourced from existing datasets or self-collected, underwent a standardized pre-processing stage to ensure consistency and quality within the Leader360V dataset. This process included video clipping, face anonymization, and other privacy-preserving operations. Additionally, we removed biased videos and balanced the distribution of different scenarios. *More details can be found in the Supplement.*

### 3.2 Automatic Annotation Pipeline

Due to the inherently large field of view (FoV), severe geometric distortion, and content discontinuities, annotating 360 video becomes particularly challenging and labor-intensive. To alleviate the burden on human annotators, we propose the Automatic Annotate Any 360 Video (A³360V) pipeline, as shown in Fig. 4, which efficiently integrates pre-trained 2D segmentors and large language models (LLMs) to streamline the labeling process. A³360V operates through a three-stage pipeline: Initial Annotation Phase (Sec. 3.2.1), Auto-Refine Annotation Phase (Sec. 3.2.2), and Manual Revise Phase (Sec. 3.2.3), which will be introduced in detail below.

### 3.2.1 Initial Annotation Phase

In the Initial Annotation Phase, given a 360 video $\mathcal{V}_i$, A³360V begins by selecting the first frame $F_1$ as the starting point for annotation. To mitigate the issue of horizontal content discontinuity caused by ERP—particularly at the left and right image borders, we first apply horizontal padding to $F_1$,

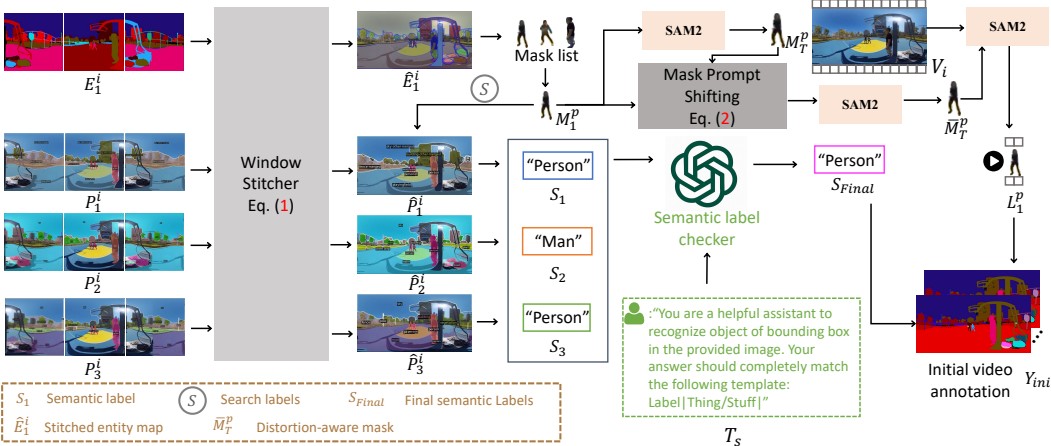

Figure 5: Illustration of the process of SDR Module.

resulting in an extended frame denoted as $F_1^p$. The $F_1^p$ is then divided into a series of overlapping patches using a horizontal sliding window. Each of these patches is subsequently processed by a diverse set of pre-trained segmentors, which we categorize into two groups: the first group comprises entity segmentors (e.g., SAM [30], CropFormer [15], E-SAM [31]), which produce class-agnostic instance-level masks capturing perceptual entities without relying on predefined taxonomies. We denote their output on frame $F_1$ as $\mathcal{E}_1^i$. The second group consists of panoptic segmentors (e.g., Mask2Former [32, 33], OneFormer [16], OMG-Seg [34]), each trained on different datasets to generate class-aware predictions. We denote one model's output on frame $F_1$ as $\mathcal{P}_1^i$. These models produce segmentation results aligned with various large label spaces (e.g., COCO [35], ADE20K [36], and Cityscapes [37]), enriching the annotation pool with complementary semantic categories.

To address the object distortion introduced by ERP and unify the semantics across heterogeneous predictions from multiple segmentors, we propose the **Semantic- and Distortion-aware Refinement (SDR) Module**, as illustrated in Fig. 5. This module plays a central role in the Initial Annotation Phase by consolidating outputs from both entity and panoptic segmentors into a coherent, distortion-aware annotation for the first frame $F_1$. While the framework supports an arbitrary number of panoptic segmentors, we illustrate our approach using three representative models in this paper. Based on the overlapping patch divisions, the SDR module first aggregates patch-wise predictions from the 2D segmentors into full-frame segmentation maps, denoted as $\mathcal{E}_1^i$, $\mathcal{P}_1^i$, $\mathcal{P}_2^i$, and $\mathcal{P}_3^i$, using a window-stitching operation defined in Eq. (1).

$$\text{Match}(M_k, M_l) = \begin{cases} 1, & \text{if IoU}(M_k, M_l) > \tau, \\ 0, & \text{otherwise} \end{cases} \tag{1}$$

where $M_k$ and $M_l$ denote instance masks predicted from overlapping regions of different patches, and $\tau$ is a predefined threshold to determine whether two masks represent the same object.

To resolve class labeling inconsistency, we incorporate a large language model (LLM)-based semantic label checker within SDR. For each entity mask proposal from $\mathcal{E}_1^i$, the pipeline retrieves corresponding label candidates from $\mathcal{P}_1^i$, $\mathcal{P}_2^i$, and $\mathcal{P}_3^i$, and feeds them into the semantic label checker via a structured prompt $T_s$. The semantic label checker selects the most semantically appropriate label, yielding a harmonized set of final semantic labels for all entities. To obtain distortion-aware masks, we leverage the robustness of video foundation models by feeding each $M_1^p$ as a mask prompt into the model in an iterative manner. Taking SAM2 [12] as an example, we first input $M_1^p$ to obtain a coarse prediction $M_T^p$. To improve its reliability under 360 distortion, we perform *Mask Prompt Shifting* by applying spatial shifts to $M_T^p$ and refeeding the shifted masks into SAM2. Since SAM2 returns a single mask per query, this process yields a set of candidate masks $\mathcal{M}_T^p = \{M_T^{p,\delta} \mid \delta \in \mathcal{D}\}$. We then select the most frequently returned result as the final refined mask $\bar{M}_T^p$:

$$\bar{M}_T^p = \arg\max_{M \in \mathcal{M}_T^p} \sum_{\delta \in \mathcal{D}} \mathbb{I}\left[\text{IoU}\left(M, \text{SAM2}(\text{Shift}(M_1^p, \delta))\right) > \tau\right], \tag{2}$$

$\mathcal{D}$ denotes the set of shift directions, $\mathbb{I}[\cdot]$ is the indicator function, and $\tau$ is the IoU threshold for mask consistency. The $\bar{M}_T^p$ is used to track the entity across subsequent frames, generating a sequence of annotations $\mathcal{L}_1^p$, which, combined with LLM-verified labels, form the initial video annotation $Y_{\text{ini}}$.

### 3.2.2 Auto-Refine Annotation Phase

Given the initial annotated frame $Y_{\text{ini}}$ produced in the previous stage, the Auto-Refine Annotation Phase aims to propagate and correct annotations across the remaining frames in the 360 video $\mathcal{V}_i$. This stage iteratively processes each frame $F_t$ using a coverage-guided strategy and performs dynamic refinement for missing or misaligned regions. At each timestamp $t$, we evaluate the coverage rate of the current annotation $Y_{\text{ini}}^t$ against a predefined threshold $\rho$. If the coverage is sufficient (i.e., coverage rate $> \rho$), we accept $Y_{\text{ini}}^t$ and use it to generate the initial annotation for the next frame, $Y_{\text{ini}}^{t+1}$. Otherwise, the **Motion-Continuity Refinement (MCR) Module** is triggered to improve the annotation quality before propagation.

To identify unannotated areas in the current frame, we employ an LLM-based agent, referred to as the **Blank Area Checker**, which is guided by a task-specific text prompt $T_B$. The prompt instructs the agent to infer the nature of each blank region. Based on this semantic inquiry, the blank region is classified into one of three types: **1. Left/Right Border Mask:** If the blank region lies near the left or right boundary of $Y_{\text{ini}}^t$, we crop and horizontally stitch the current frame $F_t$ to form a complete view of the context $F_t^l$. This operation addresses the content discontinuities inherent in ERP, allowing entity segmentors to reprocess the region with improved spatial continuity. **2. Existing Mask:** If the blank mask $M_B^i$ corresponds to a previously annotated object $M_{t-1}^a$, we invoke an LLM-based agent, referred to as the **Object Retriever**, to search for a matching mask within the prior frame's annotation $Y_{\text{ini}}^{t-1}$. If a match is successfully retrieved, the blank region inherits the same semantic label. Otherwise, it is reclassified as a new mask. **3. New Mask:** If the area represents a newly emerged object not seen in earlier frames, we treat the $M_B^i$ as a novel instance and re-enter it, together with the current frame $F_t$, into the SDR module. This process yields a refined entity segmentation and assigns an accurate semantic label. After resolving all incomplete regions, the annotated frame is updated to $Y_{\text{ref}}^t$. This refined annotation is then passed to a **VOT Model** (e.g., SAM2 [12], EdgeTAM [38], SAM2MOT[39]) for temporal smoothing and consistency adjustment. The final result is appended to the refined annotation set $Y_{\text{ref}}$, which accumulates high-quality annotations.

### 3.2.3 Manual Revise Phase

Although the Auto-Refine phase significantly reduces the need for human intervention, ensuring high-quality and consistent annotations across the entire video $\mathcal{V}_i$ still requires a final verification step. In this stage, we introduce an LLM-based agent, referred to as **Annotation Checker**, which analyzes the refined annotation $Y_{\text{ref}}$ and generates natural language modification suggestions, denoted as $C_M$. These comments highlight potential issues in spatial consistency, class accuracy, or temporal coherence. A group of human annotators then reviews and edits $Y_{\text{ref}}$ based on the LLM-generated feedback $C_M$, making targeted refinements rather than re-annotating from scratch. This human-in-the-loop revision process results in the final high-fidelity annotation set, denoted as $Y_{\text{final}}$.

## 4 Experiment

### 4.1 Implementation details

**Auto-Annotation Settings.** During the dataset construction, we employ CropFormer [15] as the entity segmentation model and OneFormer [16] as the panoptic segmentation model. Furthermore, GPT-4o [40] is incorporated as an LLM to function as a checker for semantic labels, blank areas, and annotations. *More details can be found in the Supplement.*

Table 3: Evaluation on samples of Leader360V for SAM [30] -based methods.

| Model | Leader360V-S Test | | |
| --- | --- | --- | --- |
| | $\mathcal{J}\&\mathcal{F}\uparrow$ | $\mathcal{J}\uparrow$ | $\mathcal{F}\uparrow$ |
| PerSAM[41] | 18.7 | 13.1 | 24.3 |
| SAM-PT[42] | 45.1 | 37.8 | 52.4 |
| GoodSAM[4] | 27.4 | 20.9 | 33.9 |

**Evaluation Subset.** We selected 500 videos as our sample dataset, ensuring that the distribution of scenarios and categories was similar to that of the entire dataset. Inspired by 360VOTS [10] and PanoVOS* [5], we divided the 500 videos into a training set (250), a validation set (125), and a test set (125). For the validation set and

Table 5: Qualitative comparison between VOS models for 2D video and 360 video on samples of the Leader360V dataset.

| Task | Model | Leader360V-S Test | | |
|---|---|---|---|---|
| | | $\mathcal{J}\&\mathcal{F}$ ↑ | $\mathcal{J}$ ↑ | $\mathcal{F}$ ↑ |
| 2D | XMem[44] | 42.4 | 35.9 | 48.9 |
| | AOTL[45] | 43.1 | 37.7 | 48.5 |
| | R50-AOT-L[45] | 43.9 | 39.0 | 48.8 |
| | SwinB-AOT-L (Untrained) [45] | 38.8 | 34.2 | 43.4 |
| | ☆SwinB-AOT-L (Trained) [45] | 58.3↑19.5 | 49.4↑15.2 | 67.2↑23.8 |
| 360 | GoodSAM (Untrained) [4] | 27.4 | 20.9 | 33.9 |
| | ☆GoodSAM (Trained) [4] | 60.6↑33.2 | 51.5↑30.6 | 69.7↑35.8 |

Table 6: Qualitative comparison between VOT models for 2D video and 360 video on samples of the Leader360V dataset.

| Task | Model | Leader360V-S Test | |
|---|---|---|---|
| | | $S_{dual}$ ↑ | $P_{dual}$ ↑ |
| 2D | SiamX[46] | 0.217 | 0.183 |
| | AiATrack[47] | 0.286 | 0.252 |
| | ProContEXT[48] | 0.308 | 0.270 |
| | SimTrack (Untrained)[49] | 0.291 | 0.256 |
| | ☆SimTrack (Trained)[49] | 0.417↑12.6 | 0.373↑11.7 |
| 360 | SiamX[46] | 0.282 | 0.254 |

test set, 66% of the clips are clipped from the original train set videos as val and test sets, and the rest are used as the train set. The remaining clips in the validation and test sets were selected from new and unseen scenarios.

**Evaluation Metric.** For VOS task, we choose region accuracy ($\mathcal{J}$), boundary accuracy ($\mathcal{F}$), and combined average ($\mathcal{J}\&\mathcal{F}$) as evaluation metrics, following the standard protocol [43, 5]. For VOT task, we utilize metrics of dual success ($S_{dual}$) and dual precision ($P_{dual}$), following 360VOTS [10].

## 4.2 Comparison Result Analysis

**Results via SAM-based Model.** Inspired by [5], we assess various SAM [30] versions on our Leader-360V test set, as shown in Tab. 3. Due to the domain gap between 2D and 360 images, PerSAM [41] shows poor performance. Similarly, SAM-PT [42], a SAM-based VOS model, also delivers unsatisfactory results. Additionally, GoodSAM [4], a 360 image segmentation model, is evaluated and yields disappointing outcomes. These results highlight the need for further exploration to bridge the domain gap and improve tracking performance for 360 videos.

Table 4: Comparison of results from different components.

| Phase Component | | Updated Frame | | |
|---|---|---|---|---|
| | | $\mathcal{J}\&\mathcal{F}$ ↑ | $\mathcal{J}$ ↑ | $\mathcal{F}$ ↑ |
| Phase I | 2D Segmentor | 25.1 | 16.4 | 33.8 |
| | After SDR | 67.3↑42.2 | 61.1↑44.7 | 73.5↑39.7 |
| Phase II | SAM2 | 35.8 | 26.5 | 45.1 |
| | After MCR | 75.9↑40.1 | 69.3↑42.8 | 82.5↑37.4 |

**Results of VOS Task.** We demonstrate the effectiveness of the Leader360V dataset for the VOS task in Tab. 5. While traditional 2D models show unsatisfactory performance on 360 video (e.g., XMem at **42.4** in terms of $\mathcal{J}\&\mathcal{F}$), PSCFormer, trained specifically on our train subset, exhibits significant improvement (**+36.3** for $\mathcal{J}\&\mathcal{F}$). This highlights the necessity of Leader360V for the 360VOS task.

**Results of VOT Task.** Tab. 6 presents comparison results among several trackers for both 2D and 360 tasks. Based on the quantitative results, it is evident that our dataset significantly enhances the performance of the tracker. The performance of 2D tracker SimTrack [49], which is originally designed for 2D video tasks, is obviously improved, **+12.6** for $S_{dual}$ and **+11.7** for $P_{dual}$. However, the performance of the popular 360 model [46] on our dataset does not meet our expectations.

## 4.3 Ablation Study

**Effectiveness of Phase I.** In Tab. 4, we sampled 100 frames from the dataset that necessitate auto-refinement, using the final annotations as the ground truth benchmark.

The outputs from the 2D segmentor and the semantic label checker in Phase I, are evaluated. The initial performance of the 2D segmentor is hindered by the exclusion of masks with uncertain labels, resulting in relatively low accuracy scores. Nevertheless, the SDR Module facilitates the assignment of suitable labels to previously unlabeled masks, which substantially enhances the $\mathcal{J}\&\mathcal{F}$ metric by **+42.2**. An example of mask results at various stages is depicted in Fig. 6. Upon comparison, it is evident that the 2D segmenter encounters difficulties in annotating novel objects that fall outside its distribution, and some pre-existing labels exhibit low accuracy, especially those located at a distance. The semantic label checker adeptly addresses these challenges by supplementing new labels and unifying existing labels from the our category spaces, thereby enhancing overall accuracy.

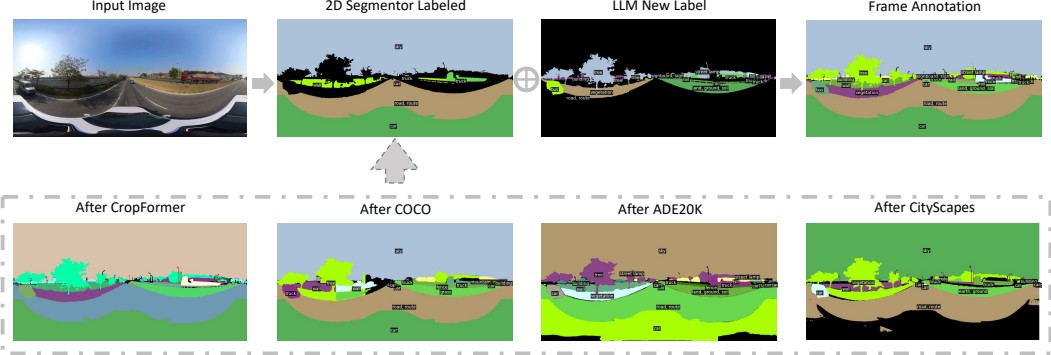

Figure 6: Example visualizations of the sequential application of entity segmentor, 2D segmentor, and semantic label checker in SDR Module.

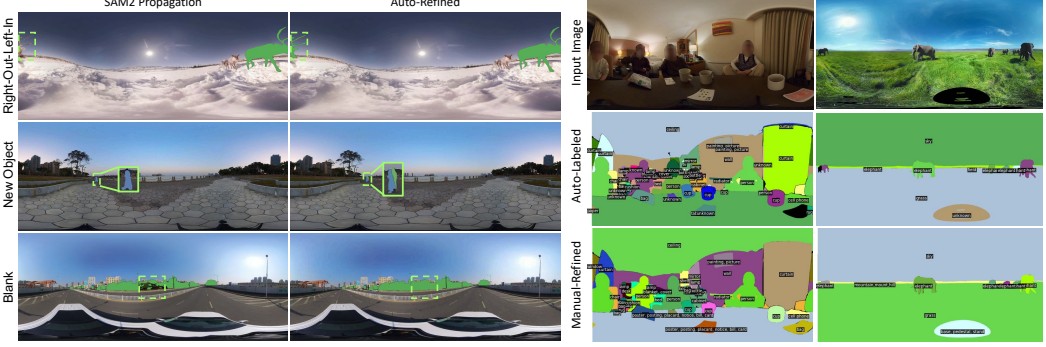

Figure 7: Example visualizations for the ablation of Auto-Refine Annotation Phase.

Figure 8: Example visualizations for the ablation of Manual Modification.

As demonstrated in Fig. 6, instances initially labeled ambiguously as "tree" and "vegetation" are ultimately unified as "tree." Additionally, the label "signboard, sign," which was overlooked by the 2D segmenter, is successfully added by the semantic label checker.

**Effectiveness of Phase II.** Tab. 4 presents a comparison of SAM2 outputs and Phase II's auto-refinement process against final annotations. SAM2 results in large blank areas for updated frames, leading to low $\mathcal{J}\&\mathcal{F}$ scores. Phase II's process increases these scores by **+40.1**, refining existing masks and adding new ones via the MCR Module. Fig. 7 illustrates three cases. The first case involves a deer that exits the right boundary of the frame and reenters from the left boundary, a scenario caused by the panorama effect. In Phase II's SDR Module, the left part of the deer is successfully segmented and assigned the same object ID as the right part, ensuring continuity. The second case illustrates a person who is obscured in the last frame but appears in the current frame. Here, the MCR segments the person and assigns a new label appropriately. The final case highlights a failure in SAM2's tracking, caused by panoramic distortion, which introduces a domain gap. The MCR Module corrects the segmentation error and restarts tracking at this frame, effectively restoring consistency.

**Effectiveness of Phase III.** The Comparison between auto-refinement masks and manual-refined masks is shown in Fig. 8. Although auto-refinement masks from our A$^3$360V pipeline demonstrate high quality, we still manually revise these masks to further enhance performance. During the revision, we specifically address issues related to object boundaries, label hallucinations, and incorrect masks.

### 4.4 Discussion

**Flexibility of A$^3$360V.** A$^3$360V's flexibility stems from its modular design, allowing users to select from various 2D segmentors for auto-annotation. For entity segmentation, options include SAM [30], CropFormer [15], and E-SAM [31], providing robust object delineation. For panoptic segmentation, models like Mask2Former [32, 33], OneFormer [16], and OMG-Seg [34] can be integrated for comprehensive scene understanding. A$^3$360V also supports the flexible selection of LLMs for label checking, ensuring compatibility with different user needs. This pipeline is versatile, applicable to both 360 and 2D videos, making it suitable for diverse video annotation tasks and adaptable to various datasets and applications. *More discussions are in the Appendix.*

# 5 Conclusion and Limitations

**Conclusion.** In this paper, we presented Leader360V, the first large-scale, labeled real-world 360 video dataset specifically designed for instance segmentation and tracking in diverse and dynamic environments. To reduce human labeling effort, we also proposed the $A^3360V$ pipeline, a three-phase framework that integrated pre-trained 2D segmentors with large language models to automate the annotation process, with minimal human intervention limited to final refinement. Extensive user studies and experimental results demonstrated the effectiveness of each stage in the pipeline and highlight the potential of Leader360V to advance research in robust, scalable 360 video understanding.

**Broader Impacts.** Leader360V has the potential to stimulate future research in 360VOT and 360VOS, and to support the development of foundation models tailored to 360 video understanding. Additionally, our $A^3360V$ pipeline offers a practical paradigm for combining large language models with pre-trained vision models to reduce manual annotation costs, which may inspire more scalable and efficient dataset construction methods in future work.

**Limitations and Future Work.** Our current Leader360V dataset does not yet cover all object classes commonly encountered in daily life, nor does it include annotations for the motion states of moving objects. In future work, we plan to further enrich the dataset annotations to support a broader range of tasks. We also aim to leverage Leader360V to explore the development of foundation models for 360 visual understanding.

**Acknowledgement.** This work is supported by the MOE AcRF Tier 1 SSHR-TG Incubator Grant FY24 (Grant No. RSTG7/24), the National Natural Science Foundation of China (Grant No. 62206069, U22B2060), the National Key R&D Program of China (Grant No. 2023YFF0725100), and the Guangdong-Hong Kong Technology Innovation Joint Funding Scheme (Project No. 2024A0505040012).

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

# A   User Study

To further investigate the effectiveness of our annotation pipeline, we randomly selected 100 videos (0.98% of the entire dataset) from Leader360V, along with 500 images randomly selected from these videos. We invited three groups of human testers with junior, average, and senior skill levels in 360 video-related tasks, each consisting of 10 members. In Fig. 9 illustrates testers' ability to distinguish between the manually revised image/video masks and the automatically annotated masks from Phase I (masks are produced by an open-source pre-trained model) or Phase II (masks are generated by our annotation pipeline, A³360V). A higher score indicates that testers can more accurately identify differences.

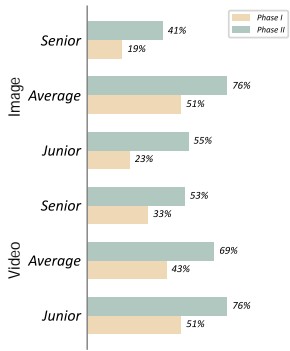

Figure 9: The user study of our Leader360V dataset.

We employed a 10-point scoring system for testers to evaluate these images and videos based on three metrics: the extent of mask missing (M), the extent of mask incorrectly annotated (W), and the annotation level on distorted objects (D). Detailed information is listed in the table of right Table 7. For **image evaluation**, a greater number of missing masks results in a lower M-score, and a higher number of incorrectly annotated masks leads to a lower W-score. Besides, the better the masks fit distorted objects, the higher the D-score. For **video evaluation**, we focused on moving objects. A greater number of missing masks for moving objects results in a lower M-score, and a higher number of incorrectly annotated masks for newly appeared objects leads to a lower W-score. Besides, the better the masks fit distorted moving objects, the higher the D-score.

Table 7: The user study of our Leader360V dataset.

| Modality | Human | Auto | | | | | | Revised | | |
| | | Phase I | | | Phase II | | | Phase III | | |
| | | M | W | D | M | W | D | M | W | D |
|---|---|---|---|---|---|---|---|---|---|---|
| **Image** | Junior | 7.9 | 7.2 | 8.4 | 8.4 | 7.9 | 9.0 | 9.7 | 9.8 | 9.7 |
| | Average | 7.5 | 6.6 | 7.3 | 9.0 | 8.6 | 9.0 | 9.3 | 9.9 | 9.7 |
| | Senior | 8.0 | 5.7 | 7.3 | 9.1 | 7.7 | 8.9 | 9.5 | 9.7 | 9.6 |
| **Video** | Junior | 5.3 | 5.1 | 6.4 | 7.4 | 7.0 | 7.9 | 9.1 | 9.8 | 9.5 |
| | Average | 4.9 | 5.0 | 6.2 | 7.4 | 7.0 | 7.9 | 9.1 | 9.8 | 9.5 |
| | Senior | 4.4 | 3.3 | 6.2 | 6.8 | 6.3 | 7.8 | 8.9 | 9.6 | 9.2 |

# B  More Discussions

## B.1  Bounding Box Annotation.

In addition to mask annotations, the Leader360V dataset includes bounding box annotations, which play a critical role in supporting various computer vision tasks such as object detection and tracking. Fig. 10 demonstrates a sample image from the dataset, highlighting how bounding box annotations accurately localize objects. The inclusion of these detailed annotations not only complements the mask annotations but also significantly broadens the dataset's applicability. This makes Leader360V a comprehensive and versatile resource, enabling researchers and developers to tackle a wide range of challenges in computer vision, particularly for 360 image and video analysis.

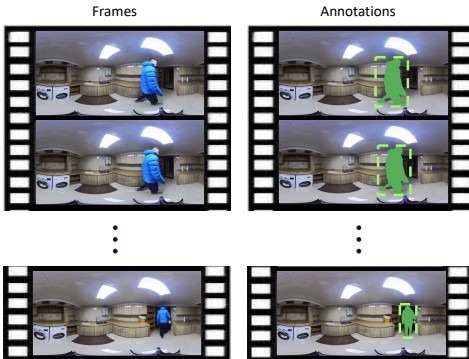

Figure 10: Samples of the bounding box annotation for VOT in the Leader360V dataset.

## B.2  Origin of Semantic Labels

Our semantic labels are derived from COCO[35], ADE20K[36], and Cityscapes[37]. During the merging process, we encountered conflicts and duplications among labels, which required careful resolution. To address this, we adhered to three key rules:

- Labels with similar or identical semantics across datasets were merged into a single unified category to ensure consistency (e.g., *building-other-merged* from COCO[35] was unified into the broader *building* category).
- Unique labels appearing in only one dataset but distinctly different from others were retained (e.g., *fountain* from ADE20K[36]).
- Rare or ambiguous subclasses (e.g., *rider*) were either removed or merged into broader categories like *person*.

Throughout this process, we prioritized maintaining the diversity of labels to ensure that the dataset remains comprehensive and effective for a wide range of segmentation tasks.

## B.3  Analysis of Annotation Cost

Annotation is an essential yet highly time-consuming process for creating pixel-level segmentation datasets. For full manual annotation, processing a single video at 1 fps can take over 20 hours. However, with our innovative $A^3$ 360V pipeline, this time is significantly reduced to just 2 hours for the same frequency, greatly improving efficiency. For automatic annotation, we utilize eight powerful 80GB H100 GPUs, which enable us to annotate 250 videos in just 2 days. The pipeline is capable of simultaneously tracking up to 63 objects, ensuring robust multi-object segmentation. This streamlined approach saves considerable time while maintaining high-quality results.

# C Licensing and Hosting

**Author Statement.** We bear all responsibilities for the licensing, distribution, and maintenance of our dataset.

**License.** Leader360V are under CC BY 4.0 license.

**Hosting.** Leader360V can be viewed and downloaded on our homepage at `https://leader360v.github.io/Leader360V_HomePage/index.html` We assure its long-term preservation for future reference and use. The annotations for questions and answers are provided in JSON file format.

**Metadata.** Metadata can be found at `https://huggingface.co/datasets/Leader360V/Leader360V`

# D Datasheet and Maintenance Plan

## D.1 Motivation

**For what purpose was the dataset created?**

**Answer:** The dataset was created to advance research in Video Object Segmentation (VOS) and Video Object Tracking (VOT) within 360 video environments. By providing panoptic annotations, it enables comprehensive pixel-level understanding of dynamic scenes, addressing unique challenges like spherical distortions and occlusions, while supporting applications in AR/VR, robotics, autonomous systems, and immersive video analysis.

**Who created the dataset (e.g., which team, research group) and on behalf of which entity (e.g., company, institution, organization)?**

**Answer:** Due to the dual-anonymous reviewing mechanism of the conference, we are unable to disclose the identities of the creators or the affiliated entity at this stage.

**Who funded the creation of the dataset?**

**Answer:** Due to the dual-anonymous reviewing mechanism of the conference, we are unable to disclose the information of the funders or the funding affiliation at this stage.

## D.2 Composition

**What do the instances that comprise the dataset represent? (e.g., documents, photos, people, countries)**

**Answer:** Each instance in the Leader360V dataset represents a video, masks within each frame, an instance ID, and a semantic label. Videos are stored in MP4 file format, while the masks, instance ID, and the semantic label are all stored in JSON format files.

**How many instances are there in total (of each type, if appropriate)?**

**Answer:** Due to the complexity of real-world scenes, it is currently challenging to provide precise instance counts. We plan to conduct a detailed statistical analysis and will share the results in future work.

**Does the dataset contain all possible instances or is it a sample (not necessarily random) of instances from a larger set?**

**Answer:** All instances in the Leader360V dataset were newly annotated by the proposed pipeline automatically and us manually. 60% of videos are from another 360 video dataset.

**Is there a label or target associated with each instance?**

**Answer:** Yes, each instance provides the masks within the corresponding frame.

**Is any information missing from individual instances?**

**Answer:** All instances are complete.

**Are relationships between individual instances made explicit (*e.g.*, users' movie ratings, social network links)?**

**Answer:** No, relationships between individual instances are not explicitly annotated in the dataset. The focus is on panoptic segmentation and tracking, with all instances treated independently within their respective scenes.

**Are there recommended data splits (*e.g.*, training, development/validation, testing)?**

**Answer:** Yes, we have provide details in our Experiment section.

**Are there any errors, sources of noise, or redundancies in the dataset?**

**Answer:** No.

**Is the dataset self-contained, or does it link to or otherwise rely on external resources (e.g., websites, tweets, other datasets)?**

**Answer:** The annotations of our dataset are annotated by our designed pipeline and human annotators. All metadata will be publicly accessible in the dataset repository. The videos are collected from websites or open-source datasets. We have taken measures such as changing the resolution, modifying the aspect ratio, changing the format, and editing to minimize the impact on the original work rights. However, it is still possible that the copyright holder may request the deletion of certain data. If this happens, we will edit the content without affecting the questions and answers. If retaining images is also not allowed, we will still keep the annotation data and provide metadata (including URL) for the corresponding images.

**Does the dataset contain data that might be considered confidential?**

**Answer:** No.

**Does the dataset contain data that, if viewed directly, might be offensive, insulting, threatening, or might otherwise cause anxiety?**

**Answer:** No.

### D.3 Collection Process

The distribution of our dataset and the details of the preprocessing are described in Sec. 3.1

### D.4 Uses

**Has the dataset been used for any tasks already?**

**Answer:** Yes, our dataset can already enhance the performance of existing models on 360 video object segmentation and tracking tasks.

**What (other) tasks could the dataset be used for?**

**Answer:** While our dataset is mainly intended for 360 video object segmentation (360VOS) and tracking (360VOT), it is also applicable to a variety of other tasks, including image-level segmentation, 360 video captioning, and video generation.

**Is there a repository that links to any or all papers or systems that use the dataset?**

**Answer:** No.

**Is there anything about the composition of the dataset or the way it was collected and preprocessed/cleaned/labeled that might impact future uses?**

**Answer:** Our dataset is constructed from a combination of existing video datasets and newly collected videos. The annotations are automatically generated using large language models and pre-trained 2D segmentors. However, it is still possible that the copyright holder may request the deletion of certain data. If this happens, we will edit the content without affecting the questions and answers. If retaining images is also not allowed, we will still keep the annotation data and provide metadata (including URL) for the corresponding images.

**Are there tasks for which the dataset should not be used?**

**Answer:** No.

### D.5 Distribution

**Will the dataset be distributed to third parties outside of the entity (e.g., company, institution, organization) on behalf of which the dataset was created?**

**Answer:** No.

**How will the dataset will be distributed (e.g., tarball on website, API, GitHub)?**

**Answer:** The code are available in `https://leader360v.github.io/Leader360V_HomePage/index.html`.

**Will the dataset be distributed under a copyright or other intellectual property (IP) license, and/or under applicable terms of use (ToU)?**

**Answer:** CC BY 4.0.

**Have any third parties imposed IP-based or other restrictions on the data associated with the instances?**

**Answer:** No.

**Do any export controls or other regulatory restrictions apply to the dataset or to individual instances?**

**Answer:** No.

### D.6 Maintenance

**Who will be supporting/hosting/maintaining the dataset?**

**Answer:** The authors will be supporting, hosting, and maintaining the dataset.

**How can the owner/curator/manager of the dataset be contacted (e.g., email address)?**

**Answer:** We will make our contact email available after the publication of the paper.

**Is there an erratum?**

**Answer:** No. We will make announcements if there are any.

**Will the dataset be updated (e.g., to correct labeling errors, add new instances, delete instances)?**

**Answer:** Yes. We will post new update in `https://leader360v.github.io/Leader360V_HomePage/index.html` or on Huggingface at `https://huggingface.co/datasets/Leader360V/Leader360V` if there is any.

**If the dataset relates to people, are there applicable limits on the retention of the data associated with the instances (e.g., were individuals in question told that their data would be retained for a fixed period of time and then deleted)?**

**Answer:** People may appear in the newly collected video. People may contact us to exclude specific data instances if they appear in the video.

**Will older versions of the dataset continue to be supported/hosted/maintained?**

**Answer:** Yes. Old versions will also be hosted in `https://huggingface.co/datasets/Leader360V/Leader360V`.

**If others want to extend/augment/build on/contribute to the dataset, is there a mechanism for them to do so?**

**Answer:** If others wish to add data, they can apply to do so provided the data is compliant and reasonable. However, making other modifications based on our dataset is currently not allowed.

