# Leader360V: A Large-scale, Real-world 360 Video Dataset for Multi-task Learning in Diverse Environments
## -Supplementary Material-

**Weiming Zhang**[1*]    **Dingwen Xiao**[1*]    **Aobotao Dai**[1*]    **Yexin Liu**[2]
**Tianbo Pan**[3]    **Shiqi Wen**[1]    **Lei Chen**[1,2]    **Lin Wang**[4† *]

[1] HKUST (GZ)    [2] HKUST    [3] National University of Singapore
[4] Nanyang Technological University

## Abstract

Due to space constraints in the main paper, we present additional details on the data analysis of Leader360V, the design of the proposed automatic annotation pipeline, and extended experimental results in the supplementary material. Specifically, in Sec. 1, we provide more details of the Leader360V dataset. In Sec. 2, we provide more details of the annotation pipeline. In Sec. 3, we provide additional benchmark comparisons and more visualizations of the annotation pipeline.

## 1 Dataset

The videos in our Leader360V dataset are sourced from two main origins: existing 360 video datasets and newly collected videos from online resources or our own recordings.

### 1.1 Videos From Existing Datasets

As shown in Tab. 1, we manually filtered and selected videos from existing 360 video datasets originally collected for various 360-related tasks:

- **360VOTS [1]** is a dedicated benchmark for 360 visual object tracking and segmentation, comprising 120 high-resolution video sequences totaling over 113K frames. It provides dense per-frame annotations and incorporates specialized representations—such as rotated bounding boxes and spherical region masks—to effectively handle challenges unique to omnidirectional content, including projection-induced distortion and left-right content discontinuities. The dataset is designed to support both tracking and segmentation tasks, with evaluation protocols and metrics tailored specifically for panoramic imagery.

- **PanoVOS [2]** is a benchmark designed for video object segmentation in 360 videos. It consists of 150 panoramic video sequences annotated with over 19,000 instance masks, generated through a human-in-the-loop process that combines manual labeling with model-assisted propagation. The dataset features diverse real-world scenarios characterized by substantial camera motion and extended temporal duration, making it a valuable resource for evaluating long-term segmentation performance under the geometric challenges of panoramic video formats.

- **WEB360 [3]** is a dataset designed to support controllable 360 video generation, consisting of approximately 2,000 panoramic video-caption pairs collected from publicly available

---

*Equal contribution. †Corresponding author.

Table 1: **Our Data Source**. **"Pct"**: percentage of selected data. **"Sel"**: specific number of selected data. VG: Video Generation. VC: Video Caption

| Source* | Task | Pct | Sel | Relabel |
|---|---|---|---|---|
| 360VOTS* [1] | 360VOT | 80% | 232 | ✓ |
| PanoVOS* [2] | 360VOS | 60% | 90 | ✓ |
| WEB360* [3] | 360 VG | 50% | 1K+ | ✓ |
| 360+x* [4] | 360 VC | 30% | 1K+ | ✓ |
| YouTube360* [5] | 360 VC | 20% | 3K+ | ✓ |
| Open Source | N/A | N/A | 1K+ | ✓ |
| Self-Collected | 360VOTS | N/A | 2K+ | ✓ |

online sources. Each video is paired with a high-quality textual description, enabling text-conditioned 360° video synthesis. By facilitating generative modeling in the panoramic domain, WEB360 provides valuable training data for data-driven synthesis and serves as a foundation for expanding 360 video research beyond traditional perception tasks.

- **360+x [4]** is a large-scale multi-modal dataset curated for comprehensive panoramic scene understanding. It contains 2,152 videos recorded across 232 real-world environments, with each scene captured simultaneously using 360 cameras and egocentric (first-person) wearable devices. The dataset is further enriched with synchronized spatial audio, action annotations, GPS metadata, and scene-level context, supporting research on cross-view alignment, audio-visual reasoning, and multi-sensory perception in 360 video settings.

- **YouTube360 [5]** focuses on generating 360 videos from conventional narrow field-of-view (FOV) inputs. The dataset consists of over 10,000 equirectangular video clips, combining the WEB360 collection with more than 8,000 additional panoramas scraped from YouTube. These clips span a wide range of scenes, including urban environments, natural landscapes, and wildlife footage. YouTube360 enables learning-based view expansion and supports VR-oriented content creation, providing diverse training data for 360 video synthesis and immersive media generation.

Since most of these datasets were not originally designed for 360 VOT or VOS tasks, we manually filtered motion-centric scenes to better align with our multi-task setting. During the selection process, we adhered to specific selection criteria: (1) Data involving scenes with dense objects, such as crowded people or stacked bicycles, were excluded, as such scenes are too complex even for SOTA methods. (2) Data that do not feature clearly moving objects, such as static footage captured in unchanging scenes, were not included. (3) Virtual videos constructed using virtual reality or simulators were also excluded. While these videos exist in existing datasets, they are challenging to classify as real-world scenarios.

In addition to those datasets not originally designed for 360 VOT or VOS tasks, as 360VOTS and PanoVOS only provide sparse annotations—typically focusing on a single target per video—we re-annotated all selected videos to include instance masks for all visible objects in each frame.

Moreover, the existing datasets exhibit imbalances in scene diversity, as shown in Fig. 1 and Fig. 2. This is evident in three key aspects: **first**, there is a significant imbalance in the ratio of indoor to outdoor videos within the dataset, as well as in the distribution of various types of labels; **second**, the ratio of dynamically shot videos to stationary ones is unbalanced; **third**, among the dynamically shot videos, the ratio of videos featuring human subjects versus those featuring vehicles (including cars, drones, etc.) is also significantly skewed. By considering the varying speeds of each vehicle, we aim to enrich the dataset's diversity, thereby mitigating the risk of overfitting. Specifically, PanoVOS [2] and 360VOTS [1] exclude labels for categories other than animals and vehicles, resulting in an uneven and unreasonable distribution of labels. Additionally, the number of outdoor videos is several times greater than that of indoor videos. However, our dataset addresses these imbalances by deliberately controlling the number of videos for each scenario.

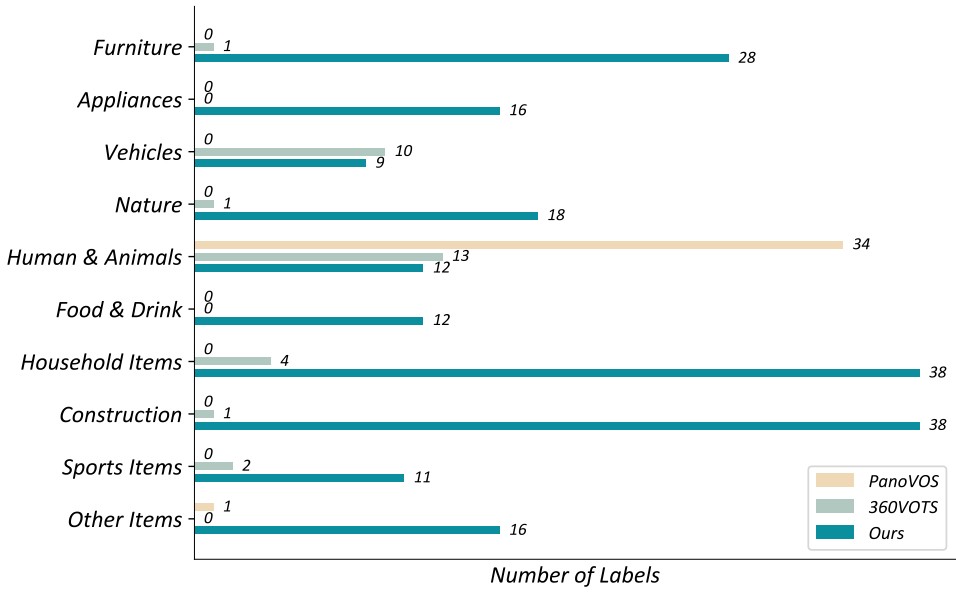

Figure 1: Comparison of category diversity among ours, PanoVOS, and 360VOTS

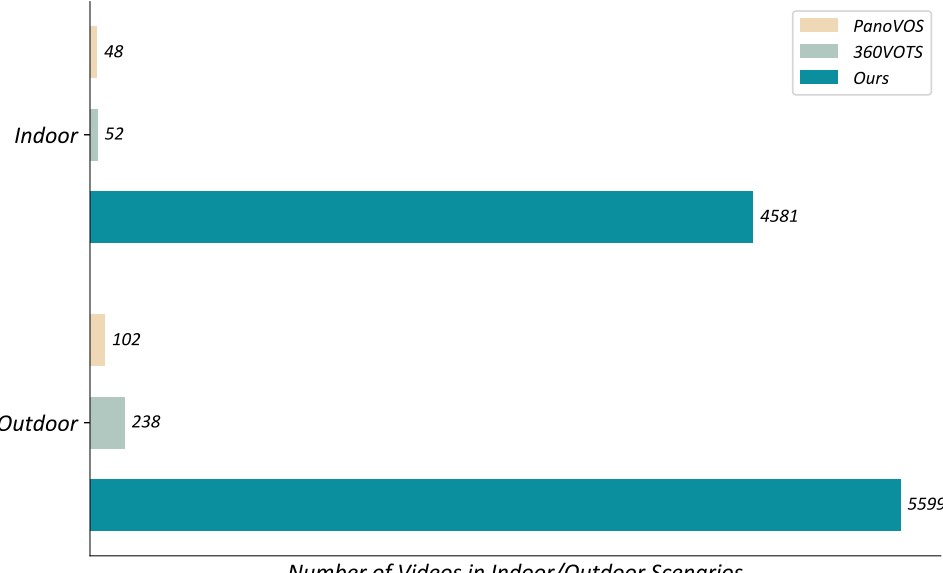

Figure 2: The overall of our Leader360V dataset.

## 1.2 Videos From Self-collected

As the videos selected from existing datasets lacked sufficient coverage of complex urban and indoor environments, we additionally collected new videos through our own recordings and publicly available online sources.

**Our new recordings.** During the data collection, we use the *Insta360 X3* camera to capture videos with at a resolution of 2048 × 1024, with a frame rate of 30 frames per second. To ensure rich scene diversity, we recorded new videos across more than ten cities, covering a wide range of both indoor and outdoor environments. Some samples from our dataset are listed in Fig. 3 and Fig. 4. Various capturing strategies were employed, including tripod-mounted stationary recording, handheld shooting with a selfie stick for walk-through scenes, and vehicle-mounted setups to simulate

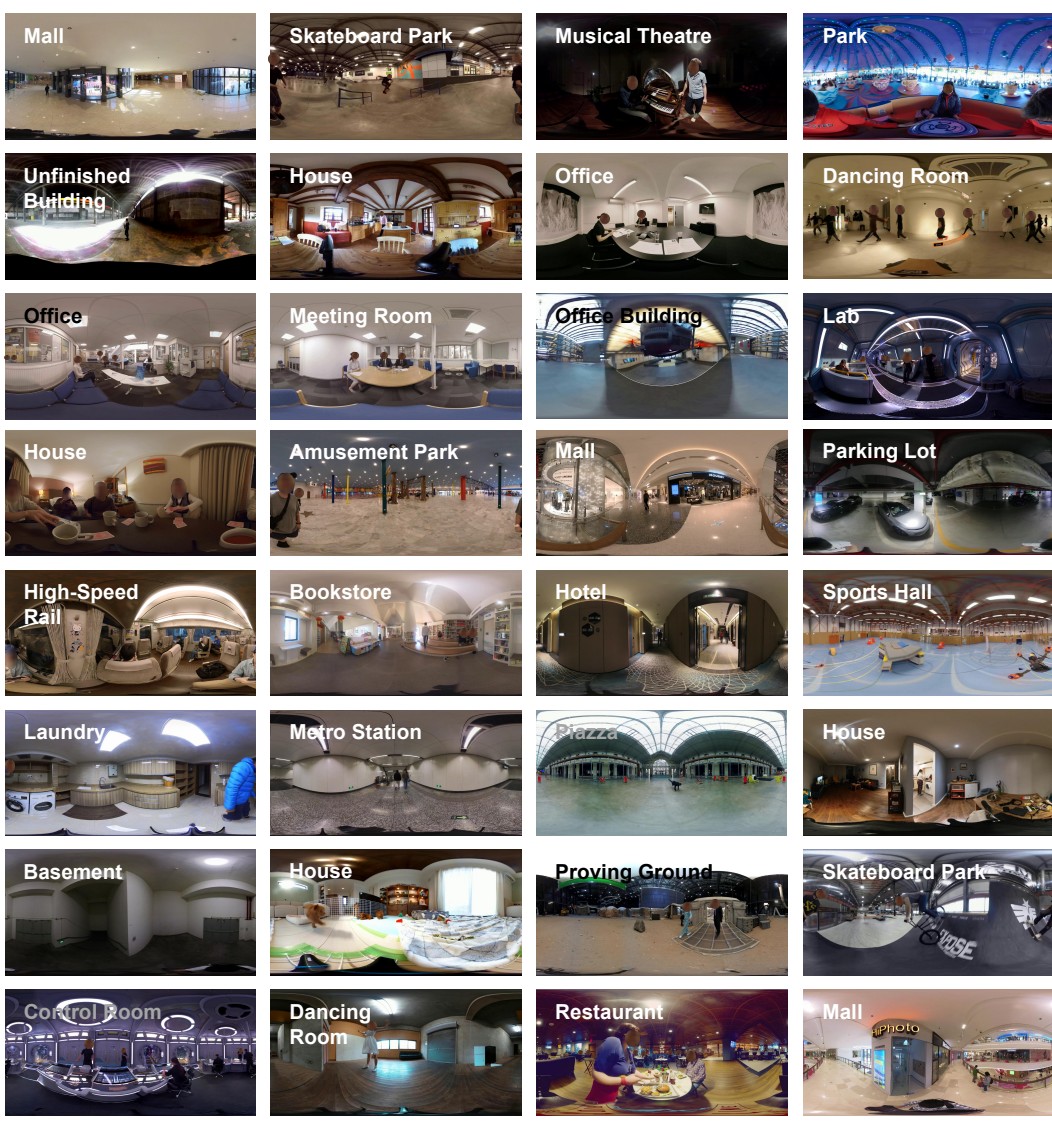

Figure 3: INDOOR Samples in Leader360V.

autonomous driving scenarios. In addition, we captured multiple views of the same scene from different angles to enhance spatial coverage.

**Online Sources.** To construct the first large-scale 360 video dataset, we additionally curated a set of newly uploaded high-resolution 360 videos from online platforms. The collection process followed four key criteria: (1) no violation of creator privacy or regional bias; (2) presence of clear object motion; (3) resolution no lower than 480p; and (4) absence of severe illumination issues, occlusions, or compression artifacts. After applying face anonymization to protect privacy, the collected videos were temporally segmented into clips with an average duration of approximately 15 seconds. In total, we successfully gathered over 1K video sequences from multiple online sources.

## 1.3 Unified Data Processing

All videos in our Leader360V, whether sourced from existing datasets or self-collected, underwent a standardized pre-processing stage to ensure consistency and quality within the Leader360V dataset. This process included video resizing ($2048 \times 1024$), video clipping, face anonymization, and other privacy-preserving operations. Additionally, we removed biased videos and balanced the distribution of different scenarios. Videos shorter than 5 seconds were excluded, and videos exceeding 30 seconds

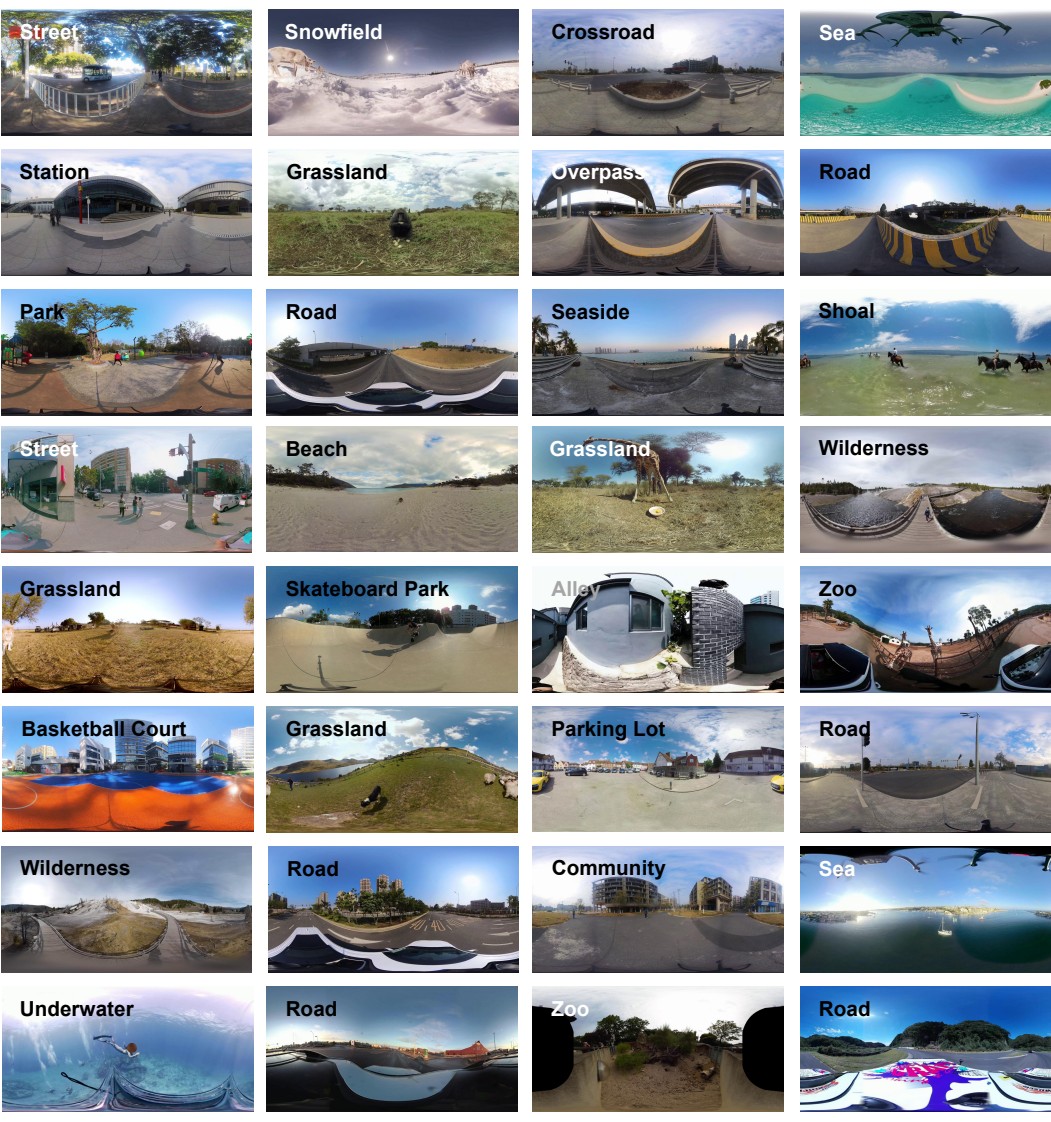

Figure 4: OUTDOOR Samples in Leader360V.

were clipped to a maximum duration of 30 seconds. The resulting videos range from 10 to 20 seconds in length, with an average duration of 15 seconds. To protect the privacy of both camera operators and passersby, we anonymized the videos by detecting faces in each frame and applying blurring filters, following a procedure similar to that described in [4].

## 2 Automatic Annotation Pipeline

In this section, we provide additional details regarding the design of our automatic annotation pipeline–$A^3$360V.

### 2.1 More Details in Initial Annotation Phase

Due to the large field-of-view (FoV) inherent in equirectangular projection (ERP), objects near the equatorial region often appear small, making them difficult to segment or track using conventional 2D segmentors or SAM2. To address this challenge, $A^3$360V first applies horizontal padding to the ERP image by cropping a portion from one side and appending it to the opposite side, thereby mitigating

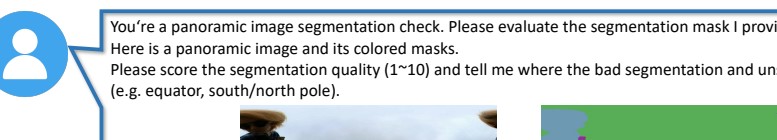

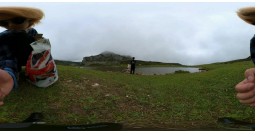
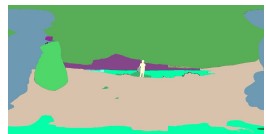

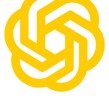

Figure 5: Visualization of an example of the feedback provided by the annotation checker, who scores the annotation and points out where bad annotations lie.

content discontinuities. The padded ERP image is then divided into three overlapping horizontal patches—left, center, and right—using a sliding window strategy.

Each patch is independently processed by 2D segmentors to produce part-wise segmentation results, denoted as $P_1^i$, $P_2^i$, and $P_3^i$, as illustrated in Fig. 5 in the main paper. We then propose a **Window Stitcher** that merges adjacent masks based on their consistency in overlapping regions and removes redundant segments. The complete algorithm is described in Alg. 1.

After obtaining the stitched segmentation results from the Window Stitcher (denoted as $\hat{P}_1^i$, $\hat{P}_2^i$, $\hat{P}_3^i$), we face the challenge of semantic inconsistency: the 2D segmentors are trained on distinct label spaces (e.g., COCO, ADE20K, Cityscapes), which may assign different labels to the same object instance. While this diversity expands the coverage of semantic predictions, it also necessitates a unified and accurate label assignment for each entity.

To address this, we introduce a large language model-based agent as the **Semantic Label Checker**, which automatically resolves label ambiguity. Specifically, for each mask, we construct a visual-language prompt by cropping the masked region from the original image and enlarging its context window. The prompt, along with candidate label names, is passed to the LLM to determine the most appropriate class name and whether the region represents a "thing" or "stuff" category. The full process is detailed in Alg. 2.

Beyond assigning a unified semantic label to each entity mask, our SDR module further refines the spatial accuracy of the mask by leveraging the strong zero-shot generalization capability of SAM2. Specifically, each verified entity mask $M_1^p$ is first input to SAM2 to obtain a coarse segmentation prediction $M_T^p$. While this initial output captures the general shape of the object, it may still be affected by the geometric distortions introduced by the ERP, particularly near the polar regions or image boundaries.

To improve robustness under such distortion, we introduce a **Mask Prompt Shifting** strategy, outlined in Alg. 3. We systematically shift the original mask prompt $M_T^p$ in four cardinal directions—up, down, left, and right—by a fixed number of pixels $\delta$, and re-input each shifted prompt to SAM2. This yields a set of candidate masks $\mathcal{M}_T^p = \{M_T^p, M_{T,u}^p, M_{T,l}^p, M_{T,r}^p, M_{T,d}^p\}$. Intuitively, if the entity is real and well-localized, the predictions from SAM2 will remain consistent across slightly perturbed prompts, due to its strong generalization to diverse mask priors.

We then select the most stable mask $\bar{M}_T^p$ based on its agreement with the majority of shifted outputs, using a voting mechanism defined by an IoU threshold (see Eq. (2) in the main paper). This procedure effectively identifies distortion-aware masks by exploiting the consistency of SAM2 responses under prompt perturbations, without requiring ground-truth supervision. It is particularly beneficial in 360 imagery, where object shapes are non-uniform and the segmentation boundaries may be misaligned due to projection artifacts.

**Algorithm 1:** Window Stitching Algorithm

**Input:** Left part predicted masks and corresponding labels: $M_l$ and $L_l$. Those for the middle part: $M_m$ and $L_m$. Those for the right part: $M_r$ and $L_r$. Padding size $w_p$ in width and overlap size $w_o$ in width. Merge IoU threshold $\tau$.

**Output:** Masks $M_f$ and labels $L_f$ for the whole 360 frame.

1   Set $w_l$ as the width of masks $M_l$, $w_m$ as the width of masks $M_m$, $w_r$ as the width of masks $M_r$.

2   360 Frame's width $w \leftarrow w_l + w_m + w_r - 2w_p - 2w_o$.

3   Initiate $M_f$ by padding $M_l$ with 0 to width $w + 2w_p$.

4   Initiate $L_f \leftarrow L_l$.

5   Set candidate mask-label pairs list $P \leftarrow [(M_m, L_m), (M_r, L_r)]$.

6   **for** *index $i$ in [0, 1]* **do**

7      Current masks and labels pair $M_c, L_c \leftarrow P[i]$.

8      **if** $i = 0$ **then**

9         Past masks in overlap part $M_{f,o} \leftarrow M_f[:,:,w_1 - w_o : w_1]$.

10     **else**

11        $M_{f,o} \leftarrow M_f[:,:,w_1 + w_2 - 2w_o : w_1 + w_2 - w_o]$.

12     Current masks in overlap part $M_{c,o} \leftarrow M_c[:,:,w_o]$.

13     $IoU_{f,c}$ is the IoU matrix of $M_{f,o}$ and $M_{c,o}$.

14     Matched index pair list $I_m$ for where $IoU_{f,c} > \tau$.

15     Pad $M_c$ with 0 to width $w + 2w_p$.

16     Initiate the removed index list $I_r \leftarrow []$.

17     **for** *index part $f_i$ and $c_i$ in $I_m$* **do**

18        **if** $L_f[f_i] = L_c[c_i]$ **then**

19           $M_f[f_i] \leftarrow M_f[f_i] \cup M_c[c_i]$.

20           Add $c_i$ to $I_r$.

21     Add masks of $M_c$ whose index not in $I_r$ to $M_f$ and labels of $L_c$ whose index not in $I_r$ to $L_f$.

22     Left pad part $M_{l,p} \leftarrow M_f[:,:,: 2w_p]$ and right pad part $M_{r,p} \leftarrow M_f[:,:,-2w_p :]$

23   $IoU_{l,r}$ is the IoU matrix of $M_{l,p}$ and $M_{r,p}$.

24   Matched index pair list $I_{l,r}$ for where $IoU_{f,c} > \tau$.

25   Set $I_r \leftarrow []$.

26   **for** *index part $l_i$ and $r_i$ in $I_{l,r}$* **do**

27     **if** $L_f[l_i] = L_f[r_i]$ *and* $l_i \neq r_i$ **then**

28        $M_f[l_i] \leftarrow M_f[l_i] \cup M_f[r_i]$.

29        Add $r_i$ to $I_r$.

30   Remove $M_f$ and $L_f$ with index in $I_r$.

31   $M_f \leftarrow M_f[:,:,w_p : -w_p]$

---

**Algorithm 2:** LLM Semantic Label Checking Algorithm

**Input:** 360 frame $F$ and one-hot mask $M$. Expand scale $\varphi$.

**Output:** Semantic label $l_M$ and $s_M$ indicates whether the entity is a stuff.

1   Masked image $F_M = F \odot M$.

2   Extract (r)BBox $B_M$ from $M$, as in [1].

3   Crop $B_M$ from $F_M$ as $F_{B,M}$.

4   Enlarge the $w$ and $h$ of $B_M$ by $\varphi$ to crop $F$ as $F_B$.

5   Input $F_{B,M}$ and $F_B$ to LLM to obtain $l_M$ and $s_M$ with system text prompt:

6   `You are a helpful assistant to recognize the object of a mask in an`
     `image. A cropped image will be provided. The known labels are classes`
     `list; do not add other new labels. Your answer should completely match`
     `the following template, no other words: Label|Thing/Stuff| If the label`
     `is not in the list, or it is hard to judge what it is, answer:`
     `Unknown|Thing| For example, bus|Thing|`

**Algorithm 3:** Mask Prompt Shifting Algorithm

---

**Input:** One-hot mask $M_1^p$. Shift scale $\delta$. IoU threshold $\tau$.
**Output:** Refined mask $M_T^p$.

1   $M_T^p \leftarrow SAM2(M_1^p)$
2   Shift $M_T^p$ up by $\delta$ pixels to get $M_{T,u}^p$.
3   Shift $M_T^p$ left by $\delta$ pixels to get $M_{T,l}^p$.
4   Shift $M_T^p$ right by $\delta$ pixels to get $M_{T,r}^p$.
5   Shift $M_T^p$ down by $\delta$ pixels to get $M_{T,d}^p$.
6   Construct candidate masks set $\mathcal{M}_T^p \leftarrow \{M_T^p, M_{T,u}^p, M_{T,l}^p, M_{T,r}^p, M_{T,d}^p\}$.
7   Extract the most stable mask $\bar{M}_T^p$ by Eq.1 in the main paper.

---

Table 2: Evaluation of domain transfer (from YouTubeVOS [6] to our Leader360V Test).

| Methods | Training Dataset | | YouTubeVOS Test | | | Leader360V Test | | |
|---|---|---|---|---|---|---|---|---|
| | YouTubeVOS | Self-Collected | $\mathcal{J}\&\mathcal{F}\uparrow$ | $\mathcal{J}\uparrow$ | $\mathcal{F}\uparrow$ | $\mathcal{J}\&\mathcal{F}\uparrow$ | $\mathcal{J}\uparrow$ | $\mathcal{F}\uparrow$ |
| **AOT**[7] | ✓ | ✗ | 72.8 | 64.2 | 84.1 | 42.9 | 37.5 | 48.3 |
| | ✓ | ✓ | 61.4↓11.4 | 53.7↓10.5 | 69.1↓15.0 | 51.6↑8.7 | 44.7↑7.2 | 58.5↑10.2 |
| **STCN** [8] | ✓ | ✗ | 75.8 | 68.9 | 82.9 | 47.2 | 38.8 | 55.6 |
| | ✓ | ✓ | 60.1↓15.7 | 51.6↓17.3 | 68.6↓14.3 | 55.4↑8.2 | 49.8↑11.0 | 61.0↑5.4 |
| **RDE**[9] | ✓ | ✗ | 61.3 | 54.9 | 67.7 | 36.2 | 30.1 | 42.3 |
| | ✓ | ✓ | 48.9↓12.4 | 41.8↓13.1 | 56.0↓11.7 | 45.5↑9.3 | 37.6↑7.5 | 53.4↑11.1 |
| **XMem**[10] | ✓ | ✗ | 76.6 | 73.3 | 79.9 | 42.5 | 35.8 | 49.2 |
| | ✓ | ✓ | 62.5↓14.1 | 57.4↓15.9 | 67.6↓12.3 | 55.1↑12.6 | 49.7↑13.9 | 60.5↑11.3 |
| **XMem++**[11] | ✓ | ✗ | 77.5 | 74.2 | 80.8 | 45.9 | 39.0 | 52.8 |
| | ✓ | ✓ | 62.7↓14.8 | 58.6↓15.6 | 66.8↓14.0 | 56.3↑10.4 | 50.6↑11.6 | 62.0↑9.2 |

## 2.2   More Details in Manual Revise Phase

In the Manual Revise Phase of the proposed $A^3360V$ pipeline, auto-annotated frames undergo a rigorous validation process via an annotation checker. This checker evaluates the accuracy and quality of the annotations, assigning a quantitative score and generating detailed modification comments, as shown in Fig. 5. These comments pinpoint areas of erroneous or suboptimal annotation, providing precise feedback on where improvements are necessary. Subsequently, human annotators can utilize image annotation tools (*i.e.*, Labelme) to refine and correct the annotations manually. This iterative process ensures high-fidelity annotations, enhancing the overall quality and reliability of the dataset for subsequent tasks.

## 3   Experiments

### 3.1   More Details in Implementation

In the implementation of our auto-annotation pipeline $A^3360V$, we utilized a series of advanced models and concise hyper-parameter settings to ensure optimal performance across various tasks. For entity segmentation, we employed CropFormer [18] with the HorNet-l [19] backbone. In the domain of panoptic segmentation, our choice was OneFormer [20] integrated with the ConvNeXt [21] backbone. For the Large Language Model (LLM), we leveraged GPT-4o [22], benefiting from its ability to understand visual and textual information and generate human-like text. Additionally, SAM2[23] was employed as the multi-object tracker, offering reliable tracking performance across diverse scenarios.

Regarding the $A^3360V$ pipeline, we meticulously set our hyperparameters to fine-tune the model's performance. The padding size $w_p$ was determined to be 128, ensuring sufficient context is maintained during processing. Similarly, the overlap size $w_o$ was also set to 128, facilitating seamless transitions

Table 3: Evaluation of domain transfer (from TrackingNet [12] to our Leader360V Test).

| Methods | Training Dataset | | TrackingNet Test | | Leader360V Test | |
|---|---|---|---|---|---|---|
| | TrackingNet | Self-Collected | $S_{dual}\uparrow$ | $P_{dual}\uparrow$ | $S_{dual}\uparrow$ | $P_{dual}\uparrow$ |
| **SiamX**[13] | ✓ | ✗ | 75.7 | 72.2 | 20.6 | 17.0 |
| | ✓ | ✓ | 60.8↓14.9 | 56.4↓15.8 | 30.9↑10.3 | 28.3↑11.3 |
| **AiATrack**[14] | ✓ | ✗ | 82.3 | 79.8 | 28.0 | 24.7 |
| | ✓ | ✓ | 64.5↓17.8 | 59.8↓20.0 | 36.1↑12.1 | 33.6↑8.9 |
| **ProContEXT**[15] | ✓ | ✗ | 84.2 | 81.8 | 30.5 | 26.9 |
| | ✓ | ✓ | 64.7↓19.5 | 59.1↓22.7 | 42.4↑11.9 | 37.9↑11.0 |
| **MixFormer**[16] | ✓ | ✗ | 83.7 | 80.9 | 29.6 | 24.4 |
| | ✓ | ✓ | 65.0↓18.7 | 60.2↓20.7 | 41.7↑12.1 | 38.3↑13.9 |
| **SimTrack**[17] | ✓ | ✗ | 83.1 | 80.6 | 29.1 | 24.0 |
| | ✓ | ✓ | 63.3↓19.8 | 58.8↓21.8 | 41.5↑12.4 | 37.6↑13.6 |

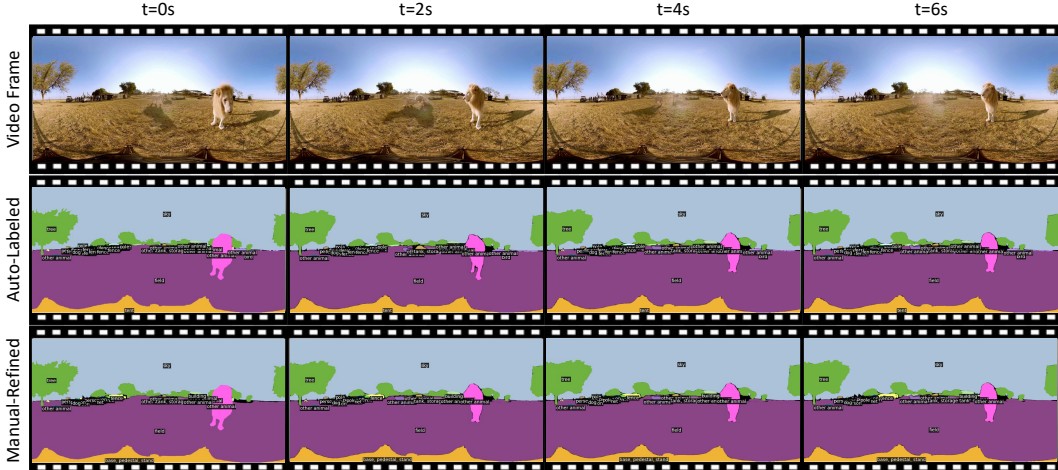

Figure 6: Visualization of auto-annotation result from A$^3$360V pipeline and manual annotation for a YouTube360 [5] video.

between segmented areas. The threshold parameter $\tau$ was configured at 0.5, optimizing the balance between precision and recall. Lastly, the bounding box enlargement size $\varphi$ was set to 100, allowing for adequate coverage and flexibility for the semantic label checker.

## 3.2 More Quantitative and Qualitative Comparisons

Tab. 2 illustrates the domain transfer results of state-of-the-art video object segmentation (VOS) models from conventional planar-domain datasets (e.g., YouTubeVOS[6]) to our panoramic Leader360V benchmark. We observe a significant performance degradation across all methods when directly applying models trained on YouTubeVOS to 360 video content. Specifically, the combined region and boundary accuracy metric $\mathcal{J}\&\mathcal{F}$ consistently drops by over 10–18 points. This highlights the substantial domain gap between conventional narrow field-of-view videos and equirectangular panoramic inputs, which introduce geometric distortion, boundary discontinuities, and changes in object appearance and motion patterns. However, after fine-tuning these models on the training split of Leader360V, we observe a notable recovery in performance. The $\mathcal{J}\&\mathcal{F}$ scores increase by 10–14 points across models, demonstrating that VOS methods can adapt effectively to the challenges of 360 scenes when provided with appropriate training data. Notably, methods like XMem [10] and XMem++ [11] exhibit strong adaptability, suggesting that memory-based and transformer-based architectures may be better suited for handling panoramic temporal dynamics. These results emphasize

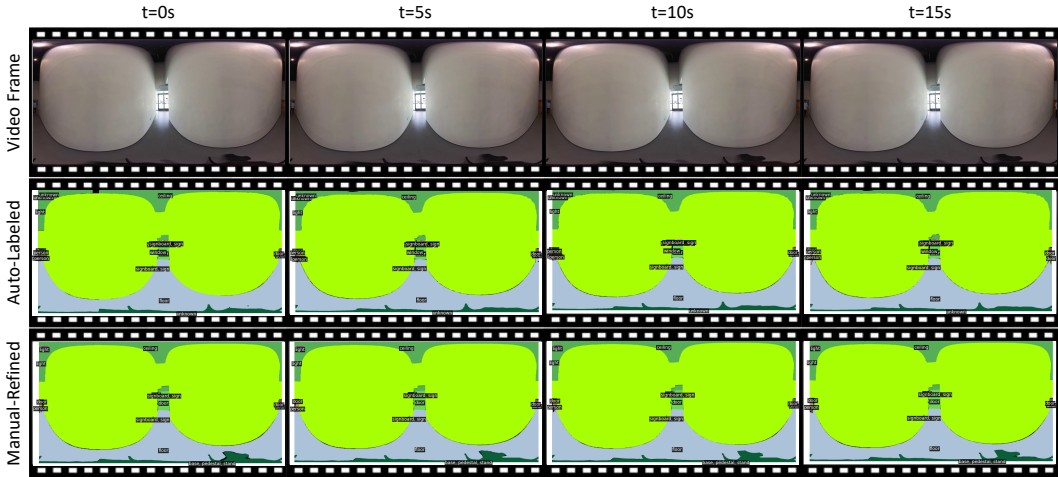

Figure 7: Visualization of auto-annotation result from A³360V pipeline and manual annotation for a self-collected indoor video.

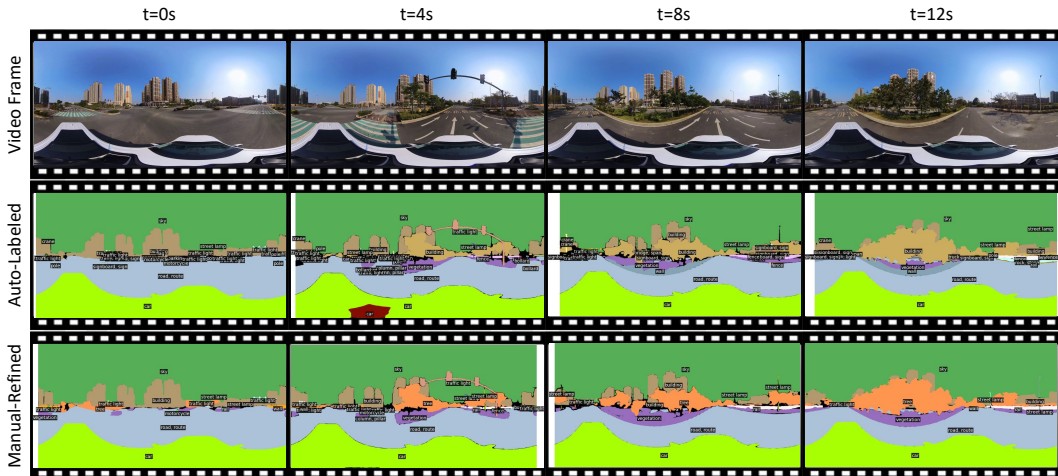

Figure 8: Visualization of auto-annotation result from A³360V pipeline and manual annotation for a self-collected outdoor video.

the importance of domain-specific training and the role of a diverse and semantically rich dataset like Leader360V in bridging the generalization gap. Additionally, the improvement indicates that label distribution mismatch—where many rare or scene-specific categories are underrepresented in planar datasets—is another contributing factor, and Leader360V's curated taxonomy helps alleviate this issue.

Tab. 3 presents the domain adaptation analysis for visual object tracking (VOT) models under similar transfer settings. Interestingly, while VOS models demonstrate reasonable recovery after fine-tuning, VOT models suffer even more pronounced performance degradation when evaluated directly on Leader360V. Performance increases of 8–14 points post-finetuning still fall short of the baseline accuracy on planar video benchmarks. This suggests that object tracking in 360 video is inherently more challenging due to compounded issues such as viewpoint wrap-around, severe distortion at the poles, and the loss of consistent object appearance across time.

These findings underscore one key insights: Leader360V serves as not only a benchmark but also an effective training resource that improves model generalization and robustness under 360 conditions. Together, these results validate our motivation to construct Leader360V and demonstrate its value in promoting research into panoramic video understanding tasks.

We also provide more visualizations about the comparison between auto-annotations and manual modifications. Fig. 6, Fig. 7, and Fig. 8 shows comparison on images from YouTube360 [5], Leader360V indoor scene, and Leader360V outdoor scene. Auto-annotation provides a quick baseline for manual labeling, and manual refinement corrects label errors and refines object boundaries. Despite needing manual adjustments, auto-annotation significantly reduces initial labeling time, making it a valuable tool for efficiently processing large datasets in VOS tasks.

### 3.3 More Discussions

**Flexibility of $A^3$360V.** A key strength of $A^3$360V lies in its high flexibility and modular design, which allows users to freely customize components based on specific annotation needs and target tasks. The framework supports interchangeable choices of 2D segmentors, large language models (LLMs), and downstream prompt strategies, enabling seamless adaptation across diverse video annotation scenarios. For 2D segmentation backbones, $A^3$360V offers configurable support for both entity-level and panoptic-level models. For entity segmentation, users may select from models such as SAM [24], CropFormer [18], and E-SAM [25], each offering strong object delineation capabilities. For panoptic segmentation, our framework integrates models like Mask2Former [26, 27], OneFormer [20], and OMG-Seg [28], each of which supports multiple semantic class taxonomies (e.g., COCO, ADE20K, Cityscapes), thereby enriching the semantic diversity and label granularity available to users. In addition, $A^3$360V supports flexible integration of LLMs for semantic label verification. The LLM-based agent can be customized via task-specific text prompts, allowing users to repurpose the same module for different roles such as label disambiguation, object categorization, or hierarchical grouping. This prompt-driven flexibility also opens opportunities for expanding $A^3$360V to other 360 video understanding tasks, such as video captioning, by simply replacing the 2D segmentors and reconfiguring the LLM agent's prompt structure. Notably, the proposed framework is not limited to 360 content. With minimal adjustments—e.g., removing ERP-specific pre-processing steps—$A^3$360V can be directly applied to large-scale 2D video datasets, enabling multi-task annotation with minimal human intervention. This generalizability makes $A^3$360V a practical and extensible tool for a wide range of video annotation applications.

**Category Superiority and Constraints.** As illustrated in Fig. 1 and Fig. 2, our Leader360V dataset encompasses a significantly larger number of semantic categories compared to existing 360VOT and 360VOS datasets. This highlights the rich semantic diversity of our annotations and the potential value of Leader360V for training more practical and generalizable 360 foundation models. While the high category count benefits from the prior knowledge embedded in pre-trained 2D segmentors and the recognition capabilities of large language models (LLMs), we do not pursue label diversity for its own sake. Excessive granularity in class definitions can lead to a highly imbalanced category distribution, where certain niche classes are represented in only a few isolated videos. To address this issue, we adopt a class merging strategy to ensure semantic coherence while preserving practical utility. For instance, in the 360+x dataset, a variety of animal-related categories are present—such as "penguin" or "gorilla"—which are rarely seen in either our newly collected videos or other public 360 datasets. As a result, we group such infrequent categories under a generalized label "other animals," while retaining commonly observed classes like "cat" and "dog" as distinct entries in our final taxonomy. Additionally, we unify synonymous or hierarchically similar categories across different segmentation taxonomies. For example, "airplane" and "plane" are merged under the label "plane"; fine-grained wall types from ADE20K such as "wall-brick," "wall-stone," "wall-tile," "wall-wood," and "wall-other-merged" are all consolidated under the label "wall." Throughout the LLM-guided annotation and final review process, we also maintain a record of novel labels proposed by the LLM agent. These candidate labels are manually reviewed and filtered before being included in the final label set. Through this class consolidation process, we arrive at a carefully curated taxonomy consisting of 198 semantic categories in Leader360V. This design balances label richness with distributional consistency, supporting both detailed scene understanding and stable model training across diverse 360 video scenarios.

**The Value of Leader360V.** As the first large-scale, real-world 360 video dataset with rich scene diversity and semantic coverage, Leader360V provides significant value for both core tasks in the 360VOTS benchmark—visual object segmentation and tracking. It addresses the current lack of well-annotated large-scale datasets in the 360 domain, which has been a major bottleneck for developing and evaluating robust models. By offering dense, high-quality annotations across a wide range of environments and object categories, Leader360V helps overcome the limitations of existing methods

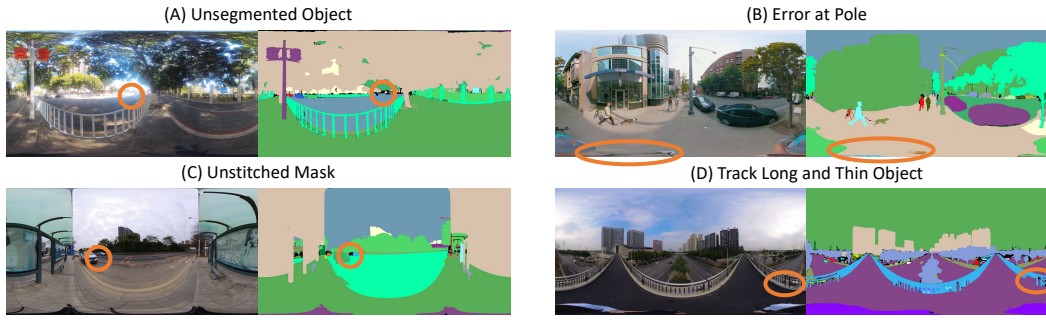

Figure 9: Visualization of A$^3$360V failure cases. Case *A* shows the unsegmented part. Case *B* is the wrong segmentation near the pole. Case *C* visualizes the failure case of window stitching. Case *D* shows disability of the employed MOT model to track long and thin objects (like rail).

in semantic recognition and scene generalization and is expected to catalyze future research progress in both 360 segmentation and tracking. Beyond task-specific benchmarks, Leader360V also fills a critical gap in the 360 community: the absence of a comprehensive dataset to support the training of large-scale segmentation and tracking foundation models. Existing approaches often rely on stitching together outputs from multiple 2D models deployed across different camera views, which incurs high computational costs and fails to capture the holistic structure of spherical scenes. In contrast, Leader360V provides a unified, native 360 representation that can support end-to-end training of 360 vision foundation models, thereby offering a scalable and efficient alternative. We believe this dataset will contribute meaningfully to the broader development of 360 video understanding in both academic and real-world applications.

**Failture Cases of A$^3$360V.** In Fig. 9, notable failure cases of A$^3$360V auto annotation pipeline are presented, underscoring challenges specific to 360 video. In the first case, the system fails to segment objects near the equator, as they are not obvious to the entity segmentation model. The second case illustrates issues occurring near the zenith or nadir of 360 images, where distortions can lead to significant segmentation errors, revealing the pipeline's limitations in handling objects around the pole. The third case is inaccurately stitching together segmented regions across the full panoramic view, causing misalignments that disrupt the coherence of the scene. The last case demonstrates the system's struggle with maintaining consistent tracking of elongated, thin objects like railings, often resulting in fragmented or missing annotations, due to the limitation of the employed MOT model (SAM2[23]). These challenges highlight the necessity for more advanced algorithms that can effectively handle the unique spatial and geometric complexities of 360 video environments, enabling more robust and accurate annotations. In future work, we plan to explore additional strategies to further enhance the performance and adaptability of our A$^3$360V framework.