# OpenReview forum: "Leader360V: A Large-scale, Real-world 360 Video Dataset for Multi-task Learning in Diverse Environment"
_NeurIPS.cc/2025/Datasets_and_Benchmarks_Track — NeurIPS 2025 Datasets and Benchmarks Track poster_

### Official Review · Reviewer_4qn9 · 2025-06-15

**Rating:** 4
**Confidence:** 4

**Summary:**

This paper presents Leader360V, a massive dataset with more than 10K annotated real-world 360° videos for multi-object tracking and instance segmentation. To annotate the dataset efficiently, the authors built the A3_360V pipeline, which uses pretrained 2D segmentors and large language models (LLMs). Experimental results indicate that models trained on this dataset achieve better performance on standard 360° scene understanding tasks.

**Dataset Code Accessibility:**

Yes

**Ethical Considerations:**

No, there are no or only very minor ethics concerns

**Final Justification:**

Keep the positive rating.

**Limitations Weaknesses:**

The database lacks some basic everyday classes and has no annotation for object motions and interactions, which restricts its suitability for dynamic scene analysis (Section 5).

There is also concern regarding dependency on GPT-4o, given that it might become unaffordable and unavailable for some users. Providing alternatives or a baseline with less complexity would serve to help.

Although automation contributes substantially to reducing labeling workload, the pipeline is complicated and needs a manual revision stage. We need to quantify how much human labour is left.

Automatically combining labels and masks could potentially introduce small errors. An error breakdown would make these limitations and improvements clear.

Urban and outdoor scenes dominate the dataset. Additional diverse geographical or scenario variation would reduce bias (Fig. 2, Section 3).

Anonymization of faces is noted, and the paper needs to make clearer statements regarding copyright and consent, particularly for online data.

**Strengths Contributions:**

Leader360V dramatically increases available material for understanding 360° video by orders of magnitude beyond existing datasets (Table 1 clearly indicates the relative scale increase).

Both segmentation and tracking annotations make it a versatile and worthwhile dataset for a variety of vision tasks.

The A3_360V automated annotation process is brilliant, utilizing pre-existing LLMs and 2D segmentors to lower the need for manual labeling. Distortion and panoramic continuity handling is especially considerate and pragmatic (see Fig. 4).

There is strong empirical evidence (user study in Table 7 and experiment results in Tables 3–5) for annotation quality and dataset efficacy.

The writing is well-structured, concise, and effectively utilizes informative graphics.

---

> ### Author Rebuttal · Authors · 2025-07-30
>
> We sincerely thank Reviewer **4qn9** for the constructive comments. We earnestly commit to refining and improving our paper based on these insightful comments, and our project code will be released to inspire more research.
>
> - **Concern about the lack of everyday classes and motion/interaction annotations**
>
> We thank the reviewer for pointing out the importance of label diversity and dynamic scene understanding. To ensure broad coverage of everyday object categories, our class list integrates labels from three widely used segmentation benchmarks: **COCO (133 classes), ADE20K (150 classes), and Cityscapes (19 classes)**. This integration enables our annotation space to encompass both common daily objects and urban/indoor semantics. Furthermore, our auto-annotation pipeline leverages multi-modal LLMs for open-world object recognition, which provides the flexibility to dynamically incorporate novel object categories beyond the predefined set. After filtering **low-confidence or overly rare labels**, we retained a final set of **198 classes (95 from COCO, 126 from ADE20K, and 14 from Cityscapes)** in our current release. Our current focus is on 360 VOT & VOS tasks. However, as our dataset is video-based and temporally continuous, it naturally supports future extensions toward modeling object motions and interactions. We are actively working on annotating motion trajectories and object-object interactions to further enhance the dataset’s utility for **dynamic scene understanding** in future releases.
>
> - **Concern about dependency on GPT-4o and model accessibility**
>
> We thank the reviewer for raising concerns regarding the accessibility and affordability of GPT-4o. To address this, we further evaluated our auto-labeling framework using a broader set of MLLMs, including models of varying sizes and capabilities. Specifically, we sampled **1000 auto-labeled frame-object pairs** and compared multiple MLLMs on two tasks: **object-labeling accuracy** and **blank-area judgment**, using human-refined annotations as the reference.
>
> **Table 4qn9-1.** Evaluation of several MLLMs on object-labeling and blank area judgment.
> | MLLM                  | Object-Labeling Accuracy$\uparrow$| Blank Area Check Accuracy$\uparrow$|
> |-----------------------|-----------------------------------|------------------------------------|
> | Without MLLM          | 0.18                              | 0.37                               |
> | GPT-4o                | 0.65                              | 0.63                               |
> | Qwen2-VL-72B          | 0.62                              | 0.66                               |
> | GLM-4.1V-9B-Thinking  | 0.48                              | 0.43                               |
> | Llama-3.2-11B-Vision  | 0.53                              | 0.58                               |
>
> As shown in **Table 4qn9-1**, while GPT-4o *(the largest model in our evaluation set)* achieves the highest object-labeling accuracy (**0.65**), smaller models such as Qwen2-VL-72B and LLaMA-3.2-Vision still perform competitively, achieving **0.62** and **0.53**, respectively. Notably, Qwen2-VL-72B even surpasses GPT-4o on blank-area accuracy (**0.66 vs. 0.63**), indicating that model size does not always correlate with performance across all tasks.
>
> Our framework is **highly flexible and model-agnostic**, allowing users to plug in alternative MLLMs depending on their compute budget or deployment constraints. While using smaller models may increase the human revision workload due to reduced precision, the pipeline remains functional and adaptable without requiring GPT-4o.
>
> - **Concern about the remaining human effort in the annotation pipeline**
>
> We appreciate the reviewer’s concern regarding the complexity of our pipeline and the remaining manual labor involved.
> As Reviewer **WrdU** emphasized, *manual correction fundamentally ensures the quality of this dataset*. To maintain this critical standard of **reliability and precision** of our annotations—especially in challenging 360° video scenes where automatic models face inherent limitations—our human revision stage remains essential.
>
> Importantly, the multi-stage design of our auto-labeling pipeline is **intentionally structured** to minimize human workload by: filtering out low-confidence outputs, refining spatial and semantic predictions, and reducing ambiguity before reaching the manual revision stage.
>
> To quantify the remaining human effort, we conducted an ablation study comparing annotation time with and without the first two stages of our pipeline.
> We randomly sampled **100 consecutive frame pairs** and measured the time cost for manual annotation after different stages in **Table 4qn9-2**.
>
> **Table 4qn9-2.** Comparison of manual annotation time cost after different stages
> | Phase                 | Time Cost (Min per Frame)$\downarrow$|
> |-----------------------|--------------------------------------|
> | Raw Frame             | 34.7                                 |
> | After Phase I         | 16.5                                 |
> | After Phase II        | 10.1                                 |
>
> It demonstrates the substantial reduction (about **70%**) in manual effort afforded by our pipeline. We would like to further optimize the process and incorporate additional metrics in future studies.
>
> - **Concern about label-mask integration limitations**
>
> We appreciate the reviewer’s suggestion and provide a breakdown of the major sources of errors observed during human revision, summarized from annotator feedback on a sample of the dataset:
>
> 1. **Semantic Label Assignment Errors (~50%):**
>  These errors occur when MLLMs assign incorrect labels to segmentation masks. Two typical cases are: (a) confusion between visually similar classes (e.g., "grass" vs. "field") due to overlapping visual cues like color and texture; and (b) misclassification as ‘unknown’ in low-confidence scenarios caused by blur, occlusion, poor lighting, or rare classes. Such ambiguity often leads the model to default to the 'unknown' label.
>
> 2. **Instance ID Association Errors (~40%):**
>  These errors arise in temporal settings, where masks of the same object instance are not consistently tracked across frames. Common issues include ID discontinuity (e.g., assigning a new ID after full occlusion) and ID swapping between similar instances (e.g., two pedestrians crossing paths), often due to limitations in Re-ID robustness and ambiguous motion cues.
>
> 3. **Mirror Reflection Misinterpretation (~10%):**
>  In scenes containing mirrors, the model sometimes segments reflected content as independent objects. However, our intended behavior is to treat the mirror surface as a single object (labeled "mirror") and ignore reflected entities. This reflects a gap between model perception and human expectations and poses a known challenge for 360° scene understanding.
>
> We are actively exploring strategies to mitigate these issues in the future, including confidence-based filtering, appearance-consistency checks, and targeted architectural refinements for reflective surfaces.
>
> - **Concern about scene diversity and dataset bias**
>
> In constructing Leader360V, we intentionally aimed to achieve a balanced representation of indoor and outdoor environments. As shown in **Figure 2 (Supplement.)**, our dataset includes **4,581 indoor videos and 5,599 outdoor videos**, indicating a relatively balanced distribution. Although the number of outdoor videos is slightly higher, the overall distribution remains relatively balanced and does not indicate a significant skew toward outdoor scenes.
>
> We will clarify this intent and distribution more explicitly in the revised manuscript to avoid misunderstanding and better emphasize our dataset's versatility.
>
> - **Concern about data anonymization, copyright, and consent clarity**
>
> We sincerely thank the reviewer for raising this important ethical concern. As documented in **Sec. 3.1.2 (anonymization)** and **Sec. D.3 (data removal)**, our pipeline implements privacy-preserving protocols that Reviewer **z2oA** confirmed *address core consent and legality considerations with no misuse concerns identified.* To augment compliance and improve traceability, we will adopt your advice and revise the paper to explicitly state our anonymization procedure, data source, and consent considerations.

---

### Official Review · Reviewer_z2oA · 2025-06-24

**Rating:** 5
**Confidence:** 4

**Summary:**

Summary
- This paper introduces Leader360V, a large-scale real-world dataset of over 10,000 360° videos with instance segmentation and multi-object tracking annotations across 198 object categories. The dataset spans diverse environments and is constructed using a novel three-stage annotation pipeline, A3360V, which integrates pre-trained 2D segmentors, large language models (LLMs), and manual revision. The dataset addresses unique challenges of 360° video such as distortion and equirectangular discontinuity. Extensive experiments demonstrate improved model performance across segmentation and tracking benchmarks when trained on Leader360V.

**Additional Feedback:**

- The idea of combining SAM2 with LLM-based label refinement is original and well-executed.
- It would be valuable to understand how the pipeline scales across more types of scene dynamics (e.g., night scenes, occlusions).
- Future extensions could explore 360-based captioning or video-language grounding using this dataset.
- Overall, the submission is technically solid and provides a much-needed benchmark for 360° visual understanding.

**Dataset Code Accessibility:**

Yes

**Dataset Code Comments:**

The dataset is hosted on Hugging Face and publicly accessible (Sec. C), and core components appear to be available for download. The paper includes links to a dataset homepage and long-term hosting plans. However, clearer dataset loading instructions and structured metadata (e.g., schema descriptions, responsible AI fields) would improve usability.

**Ethical Comments:**

The dataset contains real-world video but includes proper anonymization procedures (Sec. 3.1.2). The authors indicate that collected videos underwent privacy-preserving preprocessing and provide a mechanism for removal upon request (Sec. D.3). Although certain Responsible AI metadata (e.g., annotator demographics, data bias) are not included in structured form, related discussions are present in the paper. There are no concerns about consent, legality, or misuse at this time.

**Ethical Considerations:**

No, there are no or only very minor ethics concerns

**Final Justification:**

The authors adequately addressed my main concern by providing an ablation showing the significant impact of MLLMs on labeling accuracy. While documentation and packaging gaps remain, they are acknowledged with plans to fix in the final release. Limitations in annotation scope are reasonable for a first version. Overall, the dataset and pipeline offer strong contributions, and I maintain my score of 5 (Accept).

**Limitations Weaknesses:**

Limitations Weaknesses
- No ablation of LLM contribution: While LLMs are used for label checking and blank area analysis (Sec. 3.2.1–3.2.3), their direct contribution is not isolated or quantitatively compared to alternatives.
- Packaging and documentation gaps: Minor issues were observed in dataset packaging, such as incomplete metadata fields and one inaccessible file, according to the dataset metadata report (see Reviewer Walkthrough, Sec. 4).
- Limited annotation scope: The dataset does not include annotations for motion states or behavior-level activities, as acknowledged by the authors in Sec. 5. This limits its application in tasks like action recognition or event detection.
- No analysis of pipeline limitations: While success cases are shown (Figs. 6–8), the paper lacks examples or discussion of cases where SDR or MCR fails or underperforms.

Suggestions:
- Add an ablation experiment to isolate the effect of LLM modules in the A3360V pipeline.
- Clarify missing or incomplete elements in the dataset structure and provide improved loading documentation.
- Consider including motion/action-level annotations in future releases.
- Include qualitative examples of annotation failure modes to improve transparency.

**Strengths Contributions:**

Strengths Contributions
- Scale and diversity: The dataset is significantly larger and more diverse than existing 360° video datasets (e.g., 360VOT [13], PanoVOS [5]), as detailed in Table 1 and Figure 2.
- Task coverage: Supports both segmentation and tracking tasks with frame-level annotations, unlike prior works that are task-specific.
- Novel pipeline (A3360V): Combines multiple 2D segmentors with large language models to produce high-quality annotations efficiently.
- The Semantic- and Distortion-aware Refinement (SDR) module (Sec. 3.2.1) and Motion-Continuity Refinement (MCR) module (Sec. 3.2.2) are particularly well-motivated for handling panoramic artifacts.
- Performance impact: Experiments (Tabs. 3–5) show substantial improvements when models are trained on Leader360V versus prior datasets.
- Evaluation quality: Ablations (Sec. 4.3), qualitative visualizations (Figs. 6–8), and user studies (Sec. A) support the pipeline’s effectiveness.
- Presentation: The paper is clearly written, with well-structured sections and informative figures and tables.

---

> ### Author Rebuttal · Authors · 2025-07-30
>
> We sincerely thank Reviewer **z2oA** for the constructive comments. We earnestly commit to refining and improving our paper based on these insightful comments, and our project code will be released to inspire more research.
>
> - **Concern about the ablation of the MLLM contribution**
>
> We sincerely appreciate the reviewer for highlighting the importance of the ablation study concerning the MLLM contribution.
> The MLLM plays a crucial role as both a **semantic label checker** and a **blank area checker** during the auto-annotation phases.
> In **Table z2oA-1**, we compare several MLLMs and the pipeline without MLLM for object labeling and blank area checking.
> GPT-4o achieved the highest object-labeling accuracy (**0.65**), while the pipeline without MLLM scored **0.18**.
> For blank area checking, the pipeline without MLLM had **0.37** accuracy, whereas Qwen2-VL-72B showed a significant improvement with **0.66** accuracy.
>
> **Table z2oA-1.** Evaluation of several MLLMs on object-labeling and blank area judgment.
> | MLLM                  | Object-Labeling Accuracy$\uparrow$| Blank Area Check Accuracy$\uparrow$|
> |-----------------------|-----------------------------------|------------------------------------|
> | Without MLLM          | 0.18                              | 0.37                               |
> | GPT-4o                | 0.65                              | 0.63                               |
> | Qwen2-VL-72B          | 0.62                              | 0.66                               |
> | GLM-4.1V-9B-Thinking  | 0.48                              | 0.43                               |
> | Llama-3.2-11B-Vision  | 0.53                              | 0.58                               |
>
> It demonstrates that our flexible pipeline allows users to plug in their preferred MLLMs. Furthermore, in **Table 6 of the main paper**, we emphasize the vital roles of the SDR module and MCR module, where MLLM’s functions as the semantic label and blank area checker are pivotal.
> These experimental outcomes clearly demonstrate the significant contribution of MLLMs in: **boosting** auto-labeling accuracy and **improving** blank area checking effectiveness,  thereby substantially reducing the need for manual labeling.
> As Reviewer **4qn9** affirmed, *“The A3_360V automated annotation process is brilliant, utilizing pre-existing LLMs and 2D segmentors to lower the need for manual labeling.”*
> We will include experiments related to this ablation study in the revised version of the paper or supplementary materials.
>
> - **Concern about the packaging and documentation gaps**
>
> Because it is prohibited to update the submitted data or code repositories in the rebuttal, we apologize for the inability to update it immediately. We have identified that the errors related to metadata and inaccessibility are due to the large size of the data. Further fixes will be included in the final release, along with improved documentation to support reproducibility and usability.
>
> - **Concern about the Limited annotation scope**
>
> We thank the reviewer for this valuable suggestion. As our current annotations are designed specifically for 360 VOT&VOS tasks, they do not yet include motion states or behavior-level activities. However, our overarching goal is to build a multi-task 360 video dataset, and we see strong potential to support tasks such as **action recognition, event detection, 360-based captioning, and video-language grounding**. We will actively explore extending the annotation scope in future versions to better support these tasks.
>
> - **Concern about pipeline limitations**
>
> We appreciate the reviewer’s suggestion and provide a summary of key limitations observed in our pipeline:
>
> 1. **Semantic Misclassification:**
> While segmentation masks generally align with human perception, the model occasionally confuses visually similar classes (e.g., “grass” vs. “field”) or outputs “unknown” for rare or ambiguous objects.
>
> 2. **Fine-Grained Noise in Masks:**
> Some predicted masks contain minor artifacts—tiny polygons or degenerate shapes—which are visually negligible at standard resolution but reflect segmentation noise.
>
> 3. **Identity Tracking Failures:**
> When an object re-emerges after full occlusion, the pipeline may assign a new ID rather than preserving the original one. Identity swapping can also occur during close interactions between similar instances.
>
> 4. **Suboptimal Mirror Handling:**
> The current pipeline segments reflected content inside mirrors as separate objects. However, our intended behavior is to treat the mirror surface as a single entity labeled “mirror.” We acknowledge the challenge of modeling reflective surfaces and will explore tailored solutions in future work.
>
>
> We will include representative failure cases in the revised version to better illustrate these limitations and inform future improvement directions.

---

### Official Review · Reviewer_S7Pp · 2025-07-01

**Ethics Flags:** Safety and security, Environmental im…
**Rating:** 5
**Confidence:** 3

**Summary:**

This paper presents Leader360V, a large-scale dataset of real-world 360° videos and a novel automatic annotation pipeline that combines pre-trained 2D segmentors with large language models (LLMs) to reduce human labeling effort.

The dataset spans over 10,000 videos and nearly 200 object categories, with scenes collected from diverse sources, including cityscapes, nature, indoor scenes, and human/vehicle-based capture. The authors conduct extensive evaluations and ablation studies to validate the effectiveness of the dataset and annotation pipeline.

**Additional Feedback:**

Thank you for your detailed and courteous rebuttal. I sincerely apologize for the inaccuracies in my initial review —These were clearly a result of an inadvertent mix-up, and I appreciate your patience and professionalism in pointing them out.

I regret the oversight in my review and thank you again for handling it with grace. I remain positive about the overall quality and utility of your submission.

**Dataset Code Accessibility:**

Yes

**Dataset Code Comments:**

Can be access at hugging face and GitHub.

**Ethical Considerations:**

No, there are no or only very minor ethics concerns

**Final Justification:**

After carefully reading the authors' rebuttal, I would like to acknowledge and apologize for a mix-up in my original review that referenced unrelated aspects (e.g., 6DOF helmet-mounted data, LiDAR-derived depth, and certain baselines like YOLOv8). The authors’ response clarified that these elements are not present in the submission, and I appreciate their professional handling of the error.

Following the clarification, my evaluation focuses on the actual content of the paper, which I find to be a strong and timely dataset/system contribution. The Leader360V dataset is large-scale, diverse, and fills a major gap in the 360° video understanding community. The  annotation pipeline is thoughtfully designed, and while it primarily integrates existing tools, it does so in a well-structured and effective way that offers practical value. Experiments and user studies convincingly support the authors’ claims.

Therefore, I trend to raise my score into a 5.

**Limitations Weaknesses:**

1. While the  pipeline is well-engineered, most of its components are adaptations of existing tools — e.g., SAM2 [12], GPT-4o [40], OneFormer [16], etc. The paper focuses on system integration rather than introducing fundamentally new methods. The SDR module and MCR refinement are useful, but not algorithmically novel. (See Sec. 3.2.1–3.2.2)

2. Anothor things concerns my is the evaluation , as the scope of evaluation is restricted to a small subset, the experiments and performance metrics are mainly reported on a 500-video subset (Sec. 4.1), even though the dataset contains over 10,000 videos. This makes it hard to assess how well the improvements generalize across the full dataset. Also, the test/train/val splits come mostly from the same original sources, limiting out-of-distribution evaluation.

**Strengths Contributions:**

1. The proposed Leader360V fills a big gap in the 360° vision space — it’s the first large-scale, real-world dataset that supports both segmentation and tracking across diverse scenes. Given how limited current 360 datasets are, this is a much-needed resource that could enable new research directions.

2. The dataset covers diverse environments (urban, suburban, rural) and includes challenging lighting, occlusion, and motion scenarios. This enhances its value as a realistic benchmark for robust perception systems.

3. One big contribution of this paper is the effective use of LLMs: This paper leverages LLMs for semantic unification, gap detection, and even generating comments for human annotators. This hybrid approach is elegant.

---

> ### Author Rebuttal · Authors · 2025-07-31
>
> We sincerely thank Reviewer **S7Pp** for the time and effort spent reviewing our submission. We genuinely appreciate your service to the community and your willingness to contribute to the review process.
>
> However, we respectfully wish to seek clarification regarding several points raised in the review, as some descriptions appear to be inconsistent with the content of our manuscript. Specifically:
>
> 1. Our submission does not mention *“over 120 hours of 6DOF helmet-mounted 360° video.”*
>
> 2. We do not include or utilize *“LiDAR-derived ground truth depth”* in our dataset.
>
> 3. Our experimental section does not involve comparisons with models such as **MonoDepth2, Segment Anything, or YOLOv8**.
>
> Given these discrepancies, we are concerned that the comments may have been inadvertently associated with our submission in error. We fully understand the pressure and volume of reviewing during the submission cycle, and we deeply appreciate your work. If there has been any mix-up, we would be grateful for your help in clarifying or updating the review if needed.
>
> **Although there appear to be some inconsistencies in the review, we still appreciate the opportunity to reflect on the comments and have gained some valuable insights from your feedback：**
>
> - **Concern about the method of 360 video acquisition**
>
> We completely understand the reviewer's concerns regarding the method of 360 video acquisition. A helmet-mounted camera indeed produces data that significantly differs from vehicle-mounted sensors typically employed in standard autonomous driving datasets. However, we combine cameras with different support equipment to shoot stationary and dynamic scenes, including movements with fixed and variable frequencies, for self-collected videos. As listed in **Sec 1.2 of the Supplement**, capturing strategies include tripod-mounted stationary recording, selfie stick walk-throughs, and vehicle-mounted setups for autonomous driving simulations, are employed. This approach offers valuable diversity and realism to our dataset.
>
> - **Concern about the expansion of other 360-degree video tasks**
>
> We sincerely thank the reviewer for providing this important insight into additional depth-related and temporal tasks. Our dataset currently **focuses specifically on segmentation and tracking tasks**, and therefore, it does not incorporate depth information. However, we plan to enlarge our dataset by adding **additional videos with depth information** in future versions. Furthermore, our dataset features temporal continuity. Inspired by your feedback, we plan to improve its **applicability for temporal tasks**, such as object tracking, optical flow, and 3D trajectory prediction. On top of that, our pipeline has demonstrated remarkable flexibility, allowing us to adapt the data annotation process for various tasks effectively.
>
> - **Concern about the lack of baselines designed to exploit the unique structure of spherical data**
>
> We acknowledge the reviewer's concerns about the need for baselines tailored to spherical data; this is a valuable insight. Currently, our automatic annotation framework relies on off-the-shelf models **originally designed for 2D image segmentation**, which may not fully leverage the unique structure of spherical data. In future work, we plan to incorporate **representations that are better suited to spherical data**, such as cube maps. By training models with these more appropriate representations, we aim to significantly enhance and optimize our annotation framework, thus addressing the unique challenges and improving the overall precision and effectiveness of our pipeline.

---

> > ### Comment · Area_Chair_ooVU · 2025-08-08
> >
> > Dear S7Pp,
> >
> > The authors could not see the final rating and justification. It would be great if you could reply in here to let the authors know that whether the concerns have been resolved or not.
> >
> > Thank you.
> >
> > -AC

---

### Official Review · Reviewer_WrdU · 2025-07-02

**Rating:** 4
**Confidence:** 4

**Summary:**

The paper proposes a large-scale 360 video dataset, which contains over 10K videos. The authors designed processes for automatic annotation, refinement, and manual inspection. The proposed dataset plays a significant role in promoting the understanding of panoramic videos.

**Dataset Code Accessibility:**

Yes

**Ethical Considerations:**

No, there are no or only very minor ethics concerns

**Final Justification:**

The authors have addressed most of my concerns, and I decide to maintain my positive score.

**Limitations Weaknesses:**

1. More comparisons should be made with current 360 panoramic video datasets. For example, if the proposed dataset is used for pre-training, can fine-tuning on other datasets bring performance improvements? And what is the magnitude of the performance boost?
2. Is it reliable to use the model to verify newly emerging objects? [1] shows that current large multi-modal language models have weak object matching capabilities. Although the time interval between the two frames in this paper is relatively small, I suspect that the model may still produce some incorrect responses.

[1] Are They the Same? Exploring Visual Correspondence Shortcomings of Multimodal LLMs. Yikang Zhou, Tao Zhang, Shilin Xu, Shihao Chen, Qianyu Zhou, Yunhai Tong, Shunping Ji, Jiangning Zhang, Xiangtai Li, Lu Qi. ICCV 2025.

**Strengths Contributions:**

1. The dataset fills the gap in large-scale 360 video understanding datasets.
2. Manual correction ensures the quality of this dataset.
3. The automatic annotation process paves the way for continuous expansion.

---

> ### Author Rebuttal · Authors · 2025-07-30
>
> We sincerely thank Reviewer **WrdU** for the constructive comments. We earnestly commit to refining and improving our paper based on these insightful comments, and our project code will be released to inspire more research.
>
> - **Concern about more comparisons with 360 panoramic video datasets**
>
> We sincerely thank the reviewer for raising this important point. Due to time constraints, we sampled 400 videos from our proposed Leader360V dataset to conduct preliminary pre-training experiments. Specifically, we pre-trained two representative 360VOS models (STCN and XMem) on our dataset and then fine-tuned them on existing 360 video datasets, including **360VOS** and **PanoVOS-Test**. As acknowledged by Reviewers **z2oA** and **4qn9**, *models trained on Leader360V achieve substantial improvements versus prior datasets*. Our results align with this consensus.
>
> **Table WrdU-1.** Evaluation of models' VOS performance gain after being pretrained on Leader360 sample videos.
>
> | Model |            | 360VOS           |                |                | PanoVOS-Test           |                |                |
> |-------|------------|------------------|----------------|----------------|------------------------|----------------|----------------|
> |       |            | $J\\&F\uparrow$    | $J\uparrow$    | $F\uparrow$    | $J\\&F\uparrow$          | $J\uparrow$    | $F\uparrow$    |
> | STCN  | Untrained  | 60.9             | 55.0           | 66.7           | 50.8                   | 46.5           | 55.1           |
> |       | Pretrained | 61.6             | 55.8           | 67.3           | 52.6                   | 47.9           | 57.2           |
> | XMem  | Untrained  | 61.1             | 56.0           | 66.2           | 53.5                   | 48.3           | 58.7           |
> |       | Pretrained | 62.5             | 57.4           | 67.5           | 55.0                   | 49.8           | 60.1           |
>
> As shown in **Table WrdU-1**, both models consistently benefit from our pre-training:
> - **STCN** shows a **+1.7 and +1.8** improvement in $J\\&F$ on 360VOS and PanoVOS-Test, respectively.
> - **XMem** improves by **+1.4 and +1.5**.
>
> **Table WrdU-2.** Evaluation of models' VOT performance gain after being pretrained on Leader360 sample videos.
> | Model    |            | 360VOT            |                   |
> |----------|------------|-------------------|-------------------|
> |          |            | $S_{dual} \uparrow$| $P_{dual} \uparrow$|
> | AiATrack | Untrained  | 0.41              | 0.43              |
> |          | Pretrained | 0.42              | 0.43              |
> | SimTrack | Untrained  | 0.40              | 0.42              |
> |          | Pretrained | 0.41              | 0.43              |
>
> For the VOT task, we also pre-trained two models (AiATrack and SimTrack) on Leader360V and fine-tuned them on 360VOT in **Table WrdU-2**:
> - **AiATrack** shows **+0.01** performance gain on $S_{dual}$,
> - **SimTrack** improves **+0.01** on both  $S_{dual}$ and  $P_{dual}$,
> further corroborating Reviewer **4qn9**'s observation of "*strong empirical evidence for dataset efficacy*".
>
> These results demonstrate that our dataset serves as an effective pre-training source, providing transferable scene understanding and improving generalization to downstream 360° video datasets with limited scale.
>
> - **Concern about the reliability of newly emerging object verification**
>
> We fully agree that current large MLLMs occasionally suffer from incorrect object matching, which we also observed during the manual revision stage of our pipeline.
> To systematically evaluate this issue, we conducted an additional experiment focused on the object **grounding** capabilities of different MLLMs. Specifically, we input the object name extracted from frame $t_i$ along with an earlier frame $t_{i-10}$ into the MLLM and asked whether the object exists in the frame. We selected 50 representative video segments covering diverse challenging scenarios, such as:
>
> 1.  **Occlusion transitions** (e.g., from partially obscured to fully visible or vice versa),
> 2.  **Distance variations** (e.g., far-to-near, near-to-far),
> 3. **Boundary crossing** (a unique challenge in 360° videos), and
> 4.   **Disappearance criteria**, defined as objects becoming smaller than 400 pixels in area due to the spherical projection.
>
> **Table WrdU-3.** Comparison of several MLLMs' performance on detecting newly emerging objects.
> |        MLLM              | w/ Newly Emerging$\uparrow$| w/o Newly Emerging$\uparrow$|
> |--------------------------|----------------------------|-----------------------------|
> | GPT-4o                   | 0.8                        | 0.9                         |
> | Claude-Haiku-3           | 0.5                        | 0.7                         |
> | Qwen2.5-VL-32B-Instruct  | 0.8                        | 0.3                         |
>
> The results are summarized in **Table WrdU-3**, where the numbers indicate the success rate of correct object verification:
>
> These findings demonstrate that while **GPT-4o** performs best among the tested models, even it is not flawless in handling newly emerging objects. This further justifies the inclusion of a **manual verification stage** in our pipeline, ensuring high-quality annotations despite the inherent limitations of current MLLMs.

---

### Comment · Area_Chair_ooVU · 2025-08-03
**Reminder: Author-Reviewer Discussion (Due Aug 6 AoE)**

Dear Reviewers,

This is a friendly reminder that the author-reviewer discussion phase ends on August 6, 11:59pm AoE—around 3 days left. Please make sure to:

- Read the author responses carefully.

- Engage with authors on points they raised in the rebuttal.

- Post your initial replies as soon as possible to allow time for back-and-forth discussion.

Your timely participation is essential to ensure a fair and constructive review process. If you have any questions or need clarification, feel free to reach out.

-AC

---

### Decision · Program_Chairs · 2025-09-18

**Decision:**

Accept (poster)

**Comment:**

The final ratings of this paper are Borderline Accept, Accept, Accept, and Borderline Accept. Overall, the reviewers agree that Leader360V makes a valuable contribution: a large-scale, diverse 360° video dataset with a well-designed annotation pipeline combining 2D segmentors, LLMs, and manual revision. While there were concerns about limited methodological novelty, evaluation on only part of the dataset, lack of motion/interaction annotations, and reliance on GPT-4o. The authors’ rebuttal addressed these with additional experiments showing pre-training gains, ablations isolating LLM contributions, quantified human-effort reduction, and further clarifications. These additional experiments should be incorporated into the final version.

===== FINAL UPDATE FROM DB Track PCs ====

The final decision for this paper has been taken by the program chairs after consultation with the SACs. All Senior Area Chairs have ranked papers according to the feedback from the AC during the review process. We decided to leave the original meta-review to reflect the opinion of the AC in light of the initial discussions with reviewers and SAC.